# LEARNING IDENTIFIABLE CAUSAL STRUCTURES WITH PAIRWISE REPRESENTATION

## ABSTRACT

Supervised Causal Learning (SCL) aims to obtain causal relations from observational data, leveraging the model learned from prior datasets with ground truth causal relations. Deep Neural Network (DNN) based SCL, which learns DNNs as causal models, has gained significant attention with its numerous advantages. A recently proposed transformer-based architecture employs sample-wise and node-wise attention mechanisms to capture representations of individual variables. In the inference stage, the trained model takes the test data as input and outputs a Directed Acyclic Graph (DAG) represented as a weighted adjacency matrix.

However, this paper identifies two limitations of these approaches. First, using the adjacency matrix as a learning target can yield inconsistent results, w.r.t. structure identifiability if Bernoulli sampling is further adopted to generate the DAG. Second, current network architecture does not adequately encode the essential causal information for learning causal structures. To address these issues, we propose a novel DNN-based SCL approach, PAIRE, which incorporates a unique pairwise encoder module with a unidirectional attention layer. By taking both node features and pairwise features as layer input, it can model the internal and external relationships of variable pairs. In addition, we use a skeleton matrix along with a v-tensor, a third-order tensor representing v-structures, as our output, so as to represent the Markov Equivalence Class (MEC), which resolves identifiability inconsistency. Empirical evidence indicates PAIRE significantly outperforms other DNN-based SCL approaches.

## 1 INTRODUCTION

Supervised Causal Learning (SCL) (Dai et al., 2023; Ke et al., 2023; Ma et al., 2022) is an emerging paradigm in the field of causal discovery that seeks to learn causal relations from observational data in the following supervised setting: It has the training stage and the inference stage. In the training stage, the datasets associated with ground truth causal relations are used to learn a model to map data to the causal structure. The required training data, which comprises pairs of data and DAG, can be readily obtained via synthetic generation, or from a simulator if it is available (Lorch et al., 2022). During the inference stage, the causal structure is identified by simply applying the learned model to the target data. Typically, the causal structure is represented by a Directed Acyclic Graph (DAG) (Glymour et al., 2019).

Compared to traditional causal learning methods which treat observational data separately without supervision, the potential of SCL has been demonstrated through its strong empirical performance (Dai et al., 2023; Ma et al., 2022), as well as its robustness against sample size and distribution shift (Ke et al., 2023; Lorch et al., 2022). Notably, the deep neural network (DNN) based SCL (i.e., using DNNs to learn the mapping from data to DAG) is garnering attention due to its several advantages. These include its end-to-end training framework, which eliminates the needs of manual feature engineering, its capability to effectively handle both continuous and discrete data types, and the ability to learn latent representations.

The recent approaches CSIvA (Ke et al., 2023) and AVICI (Lorch et al., 2022) on DNN-based SCL propose a novel architecture based on transformers. This architecture incorporates sample-wise and node-wise attention mechanisms to capture the representations of individual variables, ensuring permutation invariance across samples and permutation equivariance across variables. The inference

model returns a weighted adjacency matrix $A \in \mathbb{R}^{d \times d}$, where $A_{ij} \in [0, 1]$ is the probability that $i \to j$ (indicates that $i$ is a direct cause of $j$). The final DAG $G$ is obtained as a Bernoulli sample of $A$, where each entry $G_{ij}$ is sampled from $A_{ij}$ independently. We term such approach as "Bernoulli-sampling adjacency matrix approach". Despite the encouraging results achieved thus far, we have identified two limitations in these approaches. First, the Bernoulli-sampling adjacency matrix approach can yield inconsistent results with respect to structure identifiability (identifiability for short). Second, although current attention mechanisms successfully capture sample-wise or node-wise invariance or equivariance, they do not adequately encode the essential causal information for learning causal structure. Below are the detailed elaborations:

**Risk of Bernoulli-sampling adjacency matrix approach.** One inherent characteristic of causal learning is that a DAG is only identifiable up to its Markov equivalence class (MEC) (Andersson et al., 1997; Verma & Pearl, 1990), rendering it impossible to distinguish between two DAGs within the same MEC based on available data. Considering a simulator that generates DAGs $G_1 : X \to T \to Y$ and $G_2 : X \leftarrow T \leftarrow Y$ where $X, T, Y$ are discrete random variables. As both $G_1$ and $G_2$ are within an MEC (Meek, 1995b), each observational data $D$ can be associated with either label $G_1$ or $G_2$. Even with sufficient training data generated by the simulator, if the Bernoulli-sampling adjacency matrix approach is adopted, an optimal SCL learner will learn that $0 < P_{XT} < 1$ and $0 < P_{TY} < 1$ for a test case $D$ (generated by the simulator). Consequently, there exists a non-zero probability of $P_{XT}(1 - P_{TY})$ resulting in the output of a v-structure $X \to T \leftarrow Y$, which contradicts the observed data. A concrete case study is elaborated in Sec. 4.3.

**Essential causal information.** Causal learning encompasses two fundamental tasks: the determination of the adjacency relationship between each pair of variables (skeleton learning) and the identification of the causal directions (orientation) between adjacent variables (Yu et al., 2016). The essential information required for skeleton learning is termed *persistent dependency*, indicating that a pair of variables are adjacent in the DAG if and only if they remain dependent regardless of conditioning on any subset of other variables. For orientation, an example of the essential information is termed *orientation asymmetry*. After obtaining the skeleton from the first task, we proceed to orient each unshielded triple $X - T - Y$ into a v-structure by identifying a set of variables $\mathbf{S}$, satisfying $X \perp Y \mid \mathbf{S}$ and $T \notin \mathbf{S}$. This process necessitates distinct information for the pair $\langle X, Y \rangle$ compared to the information required for the variable $T$. This asymmetry poses a challenge in encoding potential conditional dependencies of the $\langle X, Y \rangle$, which is also relevant to their persistent dependency. These essential pieces of information are not adequately encoded by current model architectures.

To address these limitations, in this paper, we propose a novel DNN-based SCL approach called PAIRE. PAIRE is equipped with a specially designed pairwise encoder module with a unidirectional attention layer. With both node features and pairwise features as the layer input, it can model both internal and external relationships of pairs of nodes. We design our learning target as a skeleton matrix together with v-tensor, a third-order tensor representing the v-structures (notably, the previously utilized adjacency matrix is a second-order tensor in comparison). These represent the MEC, which resolves the inconsistency, w.r.t. identifiability. Our contributions can be summarized as follows:

- We propose a DNN-based supervised approach, PAIRE, for causal discovery from observational data. It can learn identifiable causal structures on general discrete data and continuous data.

- We design pairwise representations modeled by a unidirectional attention layer to capture essential causal information including persistent dependency and orientation asymmetry.

- We propose to use a skeleton matrix together with a v-tensor as the model output, forming a direct representation of MEC. It resolves the inconsistency w.r.t. identifiability.

- Extensive experimental results show that PAIRE significantly outperforms other DNN-based SCL approaches. Our codes will be released for further research purposes.

## 2 RELATED WORK

The traditional methods of causal learning mostly fall into four categories: constraint-based, score-based, continuous optimization, and functional causal models. Constraint-based methods aim to

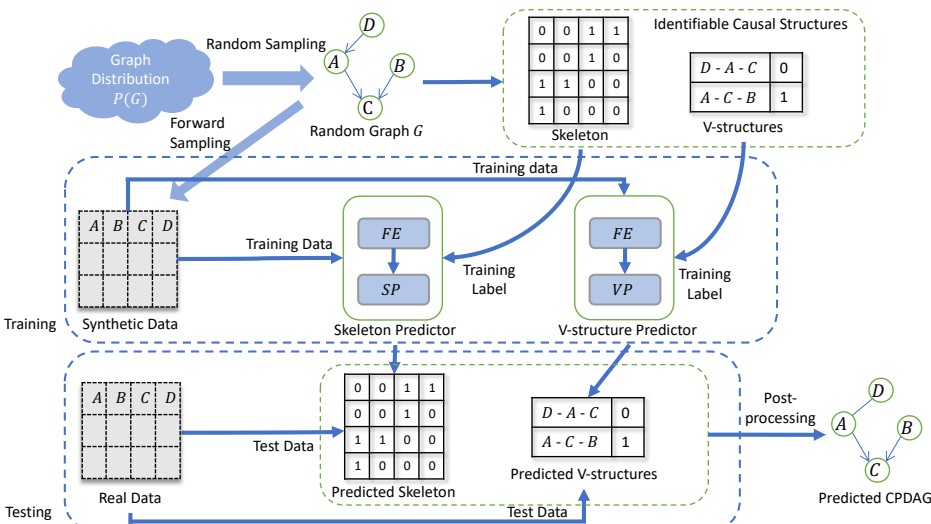

Figure 1: The whole workflow of PAIRE. During the training stage, predictors are trained to predict skeletons and v-structures with training data from sampling. During the testing stage, the predicted skeleton and v-structures are combined together to yield the final predicted CPDAG.

identify the DAG that is consistent with inter-variable conditional independence. The learning procedure of constraint-based methods first identifies the corresponding skeleton and then conducts orientation based on v-structure identification (Yu et al., 2016). The output is a CPDAG which represents the MEC. Notable algorithms in this category include PC (Spirtes et al., 2000), along with variations such as Conservative-PC (Ramsey et al., 2012), PC-stable (Colombo et al., 2014), and Parallel-PC (Le et al., 2016) designed to enhance robustness and efficiency. Score-based methods aim to find an optimal DAG according to a predefined score function, subject to combinatorial constraints. These methods employ specific optimization procedures such as forward-backward search GES (Chickering, 2002), hill-climbing (Koller & Friedman, 2009), and integer programming (Cussens, 2011). Continuous optimization methods transform the discrete search procedure into a continuous equality constraint: NOTEARS (Zheng et al., 2018) formulates the acyclic constraint as a continuous equality constraint and is further extended by DAG-GNN (Yu et al., 2019) to support non-linear causal relations. RL-BIC (Zhu et al., 2020) utilizes Reinforcement Learning to search for the optimal DAG. Recently, ENCO (Lippe et al., 2022) proposes a continuous optimization method to learn causal structure from interventional data wherein the edge existence (i.e., skeleton) and edge orientation are modeled as separate parameters. This approach aligns with the principles of our learning target design. These methods can be viewed as unsupervised since they do not access additional datasets associated with ground truth causal relations. We refer to Glymour et al. (2019); Vowels et al. (2022) for a thorough exploration of this literature.

Supervised causal learning begins from orienting edges in the context of continuous, non-linear bivariate cases under the functional causal model formalism. The task is to predict the causal direction (i.e., whether $X \rightarrow Y$ or $X \leftarrow Y$) given a set of cause-effect samples (dataset with binary labels). Supervised methods such as RCC (Lopez-Paz et al., 2015) and NCC (Lopez-Paz et al., 2017) have outperformed unsupervised approaches like ANM (Hoyer et al., 2008) or IGCI (Janzing et al., 2012) in predicting pairwise causal relations. For multivariate causal learning, ML4S (Ma et al., 2022) proposes a supervised approach specifically for skeleton learning. It employs an order-based cascade learning procedure and generates training data from vicinal graphs. Complementary to ML4S, ML4C (Dai et al., 2023) takes both data and skeleton as input and utilizes machine learning techniques to classify unshielded triples as either v-structures or non-v-structures. Although ML4S and ML4C have demonstrated impressive empirical performance, it should be noted that both methods require manual feature engineering and are only applicable to discrete data.

DNN-based SCL has emerged as a prominent approach for enabling end-to-end causal learning. Our research is primarily related to two other notable approaches, namely CSIvA (Ke et al., 2023) and AVICI (Lorch et al., 2022). They introduce novel transformer architectures that incorporate sample-

wise and node-wise attention mechanisms, enabling both permutation invariance across samples and permutation equivariance across variables. Building upon this, our approach extends the existing framework by incorporating pairwise attention to capture essential causal information more effectively, such as persistent dependency and orientation asymmetry. We further propose to use a combination of a skeleton and a set of v-structures as the learning target to address the issue of inconsistency w.r.t identifiability.

## 3 BACKGROUND

### 3.1 CAUSAL GRAPHICAL MODEL

A Causal Graphical Model is defined by a joint probability distribution $P$ over multiple random variables and a DAG $G$. Each node $X_i$ in $G$ represents a variable, and a directed edge $X_i \rightarrow X_j$ represents a causal relation from $X_i$ to $X_j$. The distribution $P$ is Markovian w.r.t. $G$, i.e., $P(X_1, X_2, \cdots, X_d) = \prod_{i=1}^{d} P(X_i \mid \mathrm{pa}_i^G)$, where $pa_i^G$ is the parent set of $X_i$ in $G$. In this work, we assume causal sufficiency, i.e., there are no latent common causes of any variables in the graph.

### 3.2 IDENTIFIABILITY

A causal DAG is in general only identifiable up to its Markov equivalence class (MEC) from observational data. The study of identifiability is well established in literature (Frydenberg, 1990; Verma & Pearl, 1990). Below we present the concepts that are relevant to the concept of identifiability. A skeleton is defined as follows:

**Definition 3.1** (Skeleton). *A skeleton $E$ defined over the data distribution $P$ is an undirected graph where an edge exists between $X_i$ and $X_j$ if and only if $X_i$ and $X_j$ are always dependent in $P$, i.e., $\forall Z \subseteq \{X_1, X_2, \cdots, X_d\} \setminus \{X_i, X_j\}$, we have $X_i \not\perp X_j | Z$.*

In this paper, we further assume $P$ is Markovian and faithful to a DAG $G$ (See details in Appendix Sec. A1.1). Therefore, the skeleton is the same as the corresponding undirected graph of the DAG $G$ (Spirtes et al., 2000). Unshielded Triples and v-structures are defined as follows:

**Definition 3.2** (Unshielded Triples (UTs) and v-structures). *A triple of variables $X, T, Y$ is an Unshield Triple (UT) denoted as $\langle X, T, Y \rangle$, if $X$ and $Y$ are both adjacent to $T$ but not adjacent to each other in the DAG $G$ or the corresponding skeleton. It becomes a v-structure denoted as $X \rightarrow T \leftarrow Y$, if the directions of the edges are from $X$ and $Y$ to $T$ in $G$.*

Formally, the Markov equivalence class is defined as follows:

**Definition 3.3** (Markov Equivalence). *Two graphs are Markov equivalent if and only if they have the same skeleton and v-structures. A Markov equivalence class (MEC) can be represented by a Completed Partially Directed Acyclic Graph (CPDAG) consisting of both directed and undirected edges. We use $CPDAG(G)$ to denote the CPDAG derived from $G$.*

According to the theorem of Markov completeness (Meek, 1995b), we can only identify a causal graph up to its MEC, i.e., the CPDAG, for discrete data or linear Gaussian data.

**Definition 3.4** (Identifiability). *Assuming $P$ is Markovian and faithful w.r.t. DAG $G$ and causal sufficiency, then each (un)directed edge in $CPDAG(G)$ indicates a (non)identifiable causal relation.*

**Remark:** The definitions of both skeleton and CPDAG are applicable for general data types, and not necessarily restricted to linear Gaussian or discrete data. So it is a clear target we can pursue. Regarding orientation, Markov completeness theorem states that for discrete or linear Gaussian data, we can only identify a causal graph up to its CPDAG; for continuous data with linear non-Gaussian mechanisms or additive noise assumptions, we can orient more causal directions (Peters et al., 2014; Shimizu et al., 2011), which is not our focus.

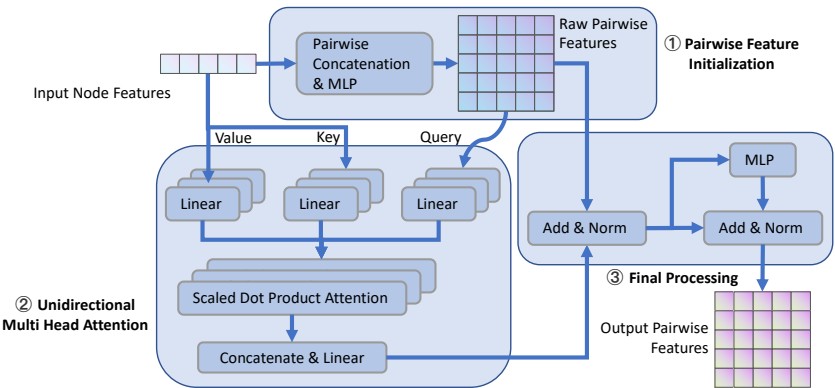

Figure 2: Illustration of the pairwise encoder module. In Part ①, it initializes raw pairwise features. In Part ②, a unidirectional attention is applied to utilized information from node features and pairwise features. In Part ③, an MLP and residual connection is used to yield final pairwise features.

## 4 METHODOLOGY

### 4.1 OVERALL WORKFLOW

The overall workflow of our PAIRE is shown in Figure 1. Each training graph $G$ is randomly sampled from a pre-defined graph meta-distribution $P(G)$, and the corresponding dataset matrix $D \in \mathbb{R}^{n \times d}$ is derived from a standard forward sampling $P(D|G)$, i.e., generating observations by sampling from the distribution defined by $G$. In our setting, we use this way to synthetically generate numerous $\langle D^i, G^i \rangle$ pairs for training.

Due to the significance of the pairwise relationship between vertices, we propose a pairwise encoder module to model the pairwise representations. Based on it, we build two neural networks whose learning targets are predicting the skeleton and the set of v-structures respectively, forming the identifiable causal structures in $G$. During inference, the skeleton and v-structure predicted by the two networks are combined together to output the final predicted CPDAG results.

### 4.2 PAIRWISE ENCODER MODULE

In this section, we present the pairwise encoder module. Given a set of $d$ nodes, $h$-dimensional node features, the goal of the pairwise encoder is to encapsulate their pairwise relationships by $d^2$ $h$-dimensional pairwise features. As shown in Sec. 1, both the internal information (i.e., the pairwise relationship) and the external information (e.g., the context of the conditional separation set) of a pair of nodes are necessary to acquire persistent dependency and orientation asymmetry. Our pairwise encoder module models the internal relationship via node feature concatenation and the non-linear mapping with MLP. Moreover, context information, including persistent dependency and orientation asymmetry, plays a crucial role in causal learning. To effectively capture such contextual relationships, the attention operation has been widely recognized as a popular approach. Therefore, we employ the attention operation within the pairwise encoder to capture these external relationships. As demonstrated in Figure 2, the pairwise encoder module consists of the following parts:

1. **Pairwise Feature Initialization.** The initial step is to concatenate the node features from a previous node feature encoder module for every pair of nodes. Subsequently, we employ a three-layer MLP to convert each $2h$-dimensional concatenated vector to an $h$-dimensional raw pairwise feature, thereby capturing the intricate relations that exist inside the pairs of nodes.

2. **Unidirectional Multi Head Attention.** In order to model the external information, we employ an attention mechanism in which the query is composed of the aforementioned $d^2$ $h$-dimensional raw pairwise features, while the keys and values consist of $h$-dimensional features of $d$ individual nodes. Note that, the attention operation in our paper is unidirectional. That is, we only calculate cross attention from raw pairwise features to individual node features. We make such a design

because both the pairwise and node information are critical to model the essential information as discussed in Sec. 1, while also maintaining a reasonable computational cost.

3. **Final Processing.** Following the widely-adopted transformer architecture, we incorporate a residual structure and dropout layer following the pairwise attention layer. Finally, we introduce a three-layer MLP to further capture intricate patterns and non-linear relationships between the input embeddings, as well as to more effectively process the information obtained from the attention mechanism. This approach allows for a comprehensive understanding of the complex relationships and leading to more robust and accurate modeling of causal structures.

Now we introduce the whole architecture of feature extractor $FE$. It consists of an input processing module, a node feature encoder module, and a pairwise encoder module sequentially. The input processing module contains a linear layer for continuous input data or an embedding layer for discrete input data. Similar to previous papers (Ke et al., 2023; Lorch et al., 2022), the node feature encoder is a transformer-like network comprising attention layers over either the observation dimension or the node dimension alternately. It naturally maintains permutation equivariance across both the variable dimension and the data dimension because of the intrinsic symmetry of attention operations. Subsequently, the pairwise encoder module is applied to obtain the pairwise features.

## 4.3 LEARNING TARGETS

### 4.3.1 CASE STUDY OF LIMITATION OF BERNOULLI-SAMPLING ADJACENCY MATRIX APPROACH

In previous work, the Bernoulli-sampling adjacency matrix approach is adopted to generate the output DAG $G$ (Ke et al., 2023; Lorch et al., 2022). In this approach, each entry $G_{ij}$ in the DAG is independently sampled from $A_{ij}$, an entry in the adjacency matrix $A$. This entry $A_{ij}$ represents the probability that $i$ directly causes $j$. We introduce a simple yet effective example setting with only three variables $X$, $Y$, and $T$ to reveal its limitation.

Considering a simulator that generates DAGs with equal probability from two DAGs: $G_1 : X \rightarrow T \rightarrow Y$ and $G_2 : X \leftarrow T \leftarrow Y$. In $G_1$, the parametrized forms are $X \sim \mathcal{N}(0, 1)$, $T = X + \mathcal{N}(0, 1)$, and $Y = T + \mathcal{N}(0, 1)$. In $G_2$, the parametrized forms are $Y = \mathcal{N}(0, 3)$, $T = \frac{2}{3}Y + \mathcal{N}(0, \frac{2}{3})$, and $X = 0.5T + \mathcal{N}(0, 0.5)$. The observational datasets coming from dags $G_1$ and $G_2$ follow the same joint distribution, which makes them inherently indistinguishable.

When using the adjacency matrix of DAG as the learning target, an optimal neural network trained with binary cross-entropy loss will predict $0.5$ probabilities on the directions of the two edges. As the prediction is regarded as a Bernoulli distribution, with $0.25$ probability the sampling result is $X \rightarrow T \leftarrow Y$. It is incompatible with the observational data, resulting in a contradictory causal structure. More details are illustrated in Figure A4 in the Appendix.

**Remark:** We clarify that our critique is specifically aimed at the limitations of the Bernoulli-sampling adjacency matrix approach, not the use of an adjacency matrix as a learning target. When the final prediction is up to an MEC rather than a fully identifiable DAG, the Bernoulli-sampling adjacency matrix approach results in inconsistency for UTs formed by non-identifiable edges. For instance, our case study shows that the entries in the adjacency matrix are not independent in determining the causal relations, thus the use of independent Bernoulli sampling over the adjacency matrix falls short of adequately representing causal relations.

### 4.3.2 LEARNING IDENTIFIABLE CAUSAL STRUCTURES

To address the issue, we propose to allow the network model to learn solely the identifiable causal structures in $G$, i.e., its MEC. As shown in Sec. 3, an MEC can be represented by a combination of the skeleton and the set of v-structures, which are our learning targets.

**Skeleton Prediction.** As the persistent dependency between pairs of nodes determines the existence of edges in the skeleton, each pairwise feature corresponds to an edge in the skeleton naturally. Therefore, for the skeleton learning task, we initially employ a max-pooling layer over the observation dimension to obtain an $h$-dimensional vector for each pair of nodes. Then, a linear layer is applied to map the pairwise features to the final prediction of edges. Our learning label, the

Table 1: Skeleton prediction results on linear Gaussian and general nonlinear continuous data. "*" indicates the case is considered failed as the algorithm takes more than 24 hours per graph.

| Graph Type | Method | Linear Gaussian | | General nonlinear | |
|---|---|---|---|---|---|
| | | F1 | Acc. | F1 | Acc. |
| WS | PC | 0.304 | 0.656 | 0.361 | 0.699 |
| | GES | * | * | 0.417 | 0.666 |
| | NOTEARS | 0.333 | 0.651 | 0.376 | 0.646 |
| | DAG-GNN | 0.355 | 0.554 | 0.323 | 0.644 |
| | AVICI | $0.446 \pm 0.013$ | $0.734 \pm 0.003$ | $\mathbf{0.536 \pm 0.008}$ | $0.728 \pm 0.002$ |
| | PAIRE | $\mathbf{0.479 \pm 0.015}$ | $\mathbf{0.750 \pm 0.003}$ | $0.500 \pm 0.037$ | $\mathbf{0.771 \pm 0.004}$ |
| SBM | PC | 0.588 | 0.900 | 0.575 | 0.893 |
| | GES | 0.708 | 0.894 | 0.565 | 0.849 |
| | NOTEARS | 0.801 | 0.945 | 0.556 | 0.861 |
| | DAG-GNN | 0.662 | 0.874 | 0.471 | 0.821 |
| | AVICI | $0.836 \pm 0.004$ | $0.959 \pm 0.001$ | $0.739 \pm 0.000$ | $0.937 \pm 0.001$ |
| | PAIRE | $\mathbf{0.853 \pm 0.007}$ | $\mathbf{0.962 \pm 0.003}$ | $\mathbf{0.809 \pm 0.004}$ | $\mathbf{0.954 \pm 0.001}$ |

undirected graph representing the skeleton, can be easily calculated by summing the adjacency of the DAG and its transpose. Denoting the combination of max-pooling and linear layer as a skeleton prediction module $SP$, our learning target for the skeleton prediction task can be formulated as $\min_{\theta,\xi} \mathcal{L}(SP_{\theta}(FE_{\xi}(D)), G + G^T)$, where $\mathcal{L}$ is the popularly used binary cross-entropy loss function, $FE$ is the feature extractor mentioned above, and $D$ denotes the input data.

**V-structure Prediction.** A UT $\langle X, T, Y \rangle$ is a v-structure when $\exists \mathbf{S}$, such that $T \notin \mathbf{S}$ and $X \perp Y | \mathbf{S}$. Therefore, we concatenate the corresponding pairwise features of the pair $\langle X, Y \rangle$ with the node features of $T$ as the feature for each UT $\langle X, T, Y \rangle$. After that, we use a three-layer MLP to predict the existence of v-structures among all UTs. For the given dataset with $d$ nodes, it outputs a third-order tensor of shape $\mathbb{R}^{d \times d \times d}$, namely v-tensor, corresponding to the predictions of the existence of v-structures. The v-tensor label can be obtained by $V_{ijk} = G_{ji}G_{ki}(1 - G_{jk})$, where $V_{ijk}$ indicates the existence of v-structure $X_j \rightarrow X_i \leftarrow X_k$. Denoting the v-structure prediction module as $VP$, the learning target for the v-structure prediction task can be formulated as $\min_{\phi,\xi} \mathcal{L}_{UT}(VP_{\phi}(FE_{\xi}(D)), V)$, where $\mathcal{L}_{UT}$ means the binary cross-entropy loss is masked by UTs, i.e., we only calculate such loss on the valid UTs. Note that the parameters $\xi$ of $FE$ are initialized from the skeleton prediction task, as the UTs to be classified are obtained from the predicted skeleton and the skeleton prediction can be seen as a general pre-trained task.

It is noteworthy that neural network models have a theoretical guarantee of the asymptotic correctness with respect to the sample size on predicting skeleton and v-structures. Formally, we have

**Theorem 4.1.** *Under canonical assumption (Definition A1.2) and the assumption that neural network can be used as a universal approximator (Assumption A1.4), there exists neural network models that always predict the correct skeleton and v-structures with sufficient samples in $D$.*

The proof and relevant discussions are provided in Appendix Sec. A1.

## 5 EXPERIMENTS

In this section, we report on a series of experimental results. Our evaluation part is mostly about OOD setting, i.e., the distribution of test set is OOD w.r.t. the distribution of training set. Specifically, our distribution of training set is generated by Erdos-Rényi (ER) and Scale-Free (SF) mechanisms, but we test our model on Watts-Strogatz (WS), Stochastic Block Model (SBM), and one real-world dataset Sachs (Sachs et al., 2005), which are with significantly different generating mechanisms. F1-scores, accuracy and SHD are general metrics for traditional methods and DNN-based methods. However, for DNN-based SCL method, AUC and AUPRC are more reasonable metrics because they avoid the influence of threshold selection. More detailed experimental settings are provided in Appendix A3. Extra experimental results are included in Appendix A4.

Table 2: CPDAG prediction results on linear Gaussian data. "*" indicates the case is considered as failed as the algorithm takes more than 24 hours per graph.

| Graph Type | Method | V-structure F1 | Identifiable edges F1 | SHD |
|---|---|---|---|---|
| WS | PC | 0.156 | 0.160 | 170.36 |
| | GES | * | * | * |
| | NOTEARS | 0.279 | 0.315 | 159.82 |
| | DAG-GNN | **0.322** | 0.327 | 193.7 |
| | AVICI | 0.277 | 0.356 | 117.62 |
| | **PAIRE** | $0.298 \pm 0.076$ | $\mathbf{0.370 \pm 0.062}$ | $\mathbf{116.797 \pm 7.253}$ |
| SBM | PC | 0.349 | 0.359 | 56.44 |
| | GES | 0.539 | 0.550 | 60.30 |
| | NOTEARS | 0.762 | 0.778 | 26.70 |
| | DAG-GNN | 0.603 | 0.625 | 61.04 |
| | AVICI | 0.792 | 0.818 | 17.48 |
| | **PAIRE** | $\mathbf{0.805 \pm 0.016}$ | $\mathbf{0.826 \pm 0.029}$ | $17.147 \pm 1.573$ |

## 5.1 GENERAL PERFORMANCE

**Skeleton Prediction.** We conduct a comprehensive comparison of PAIRE with various baseline algorithms in the skeleton prediction task. We perform experiments in Table 1 and Appendix Table A7 on both linear Gaussian and general nonlinear datasets to evaluate the performance of the competing methods. Our findings reveal that DNN-based SCL methods, i.e., AVICI and PAIRE, demonstrate a marked advantage over traditional approaches in the skeleton prediction task. Among all the methods evaluated, our proposed PAIRE emerges as the top performer, further substantiating its superiority in addressing the causal learning problem.

**CPDAG Prediction.** Table 2 and Appendix Table A8 present the evaluation outcomes of various methods applied to the CPDAG prediction task. We present our result of CPDAG prediction on the linear Gaussian data, as CPDAG reflects the maximum number of causal directions that can be oriented in this setting (Meek, 1995b). A key observation from the results is that DNN-based SCL approaches still exhibit superior performance compared to classical algorithms across all types of graphs. Our proposed PAIRE method outperforms all other methods, further validating the effectiveness of our proposals and the inherent advantages of PAIRE.

Table 3: AUC and AUPRC of skeleton prediction results on Sachs dataset.

| Method | AUC | AUPRC |
|---|---|---|
| AVICI | 0.563 | 0.405 |
| PAIRE | **0.631** | **0.468** |

## 5.2 PERFORMANCE ON REAL DATASET

To assess the practical applicability of PAIRE, we conduct a comprehensive comparison using a real-world dataset presented by Sachs et al. (2005) and available in the Bnlearn repository (Scutari, 2010). It comprises discretized measurements of 11 proteins involved in human immune system cells, providing a valuable benchmark for evaluating the performance of our approach. The DNN-based methods are trained on random SF graphs, making this an out-of-distribution prediction task.

Interestingly, the DAG label of the Sachs dataset does contains no v-structure, implying a low number of predicted v-structures is desirable. As shown in the skele-

Table 4: CPDAG prediction results on Sachs dataset.

| Method | SHD | Num. v-struc. |
|---|---|---|
| PC | 19 | 19 |
| GES | 19 | 11 |
| DAG-GNN | 13 | **0** |
| AVICI | 13 | 3 |
| PAIRE | **10** | **0** |

ton prediction results given in Table 3, PAIRE performs best under the comparisons using AUC and AUPRC. Moreover, on the final CPDAG prediction tasks shown in Table 4, PAIRE predicts no v-structures, which still surpasses other methods. This finding further substantiates the efficacy of PAIRE and emphasizes the importance of respecting identifiability.

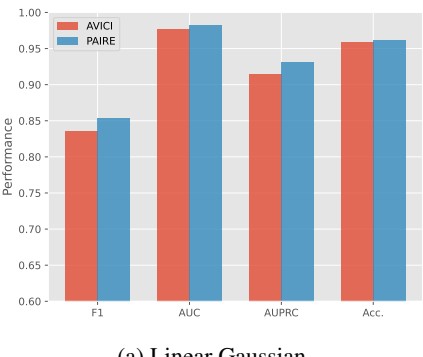

(a) Linear Gaussian

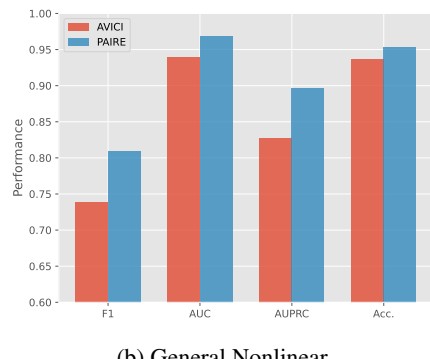

(b) General Nonlinear

Figure 3: Comparison between AVICI and PAIRE on skeleton prediction task.

Table 5: Comparison of CPDAG results from models with different learning targets.

| Graph Type | Learning Target | V-structure F1 | Identifiable edges F1 | SHD |
|---|---|---|---|---|
| WS | Adjacency Matrix | 0.257 | 0.338 | 117.65 |
| | Skeleton + V-tensor | **0.298 ± 0.076** | **0.370 ± 0.062** | **116.80 ± 7.25** |
| SBM | Adjacency Matrix | 0.778 | 0.806 | 18.31 |
| | Skeleton + V-tensor | **0.805 ± 0.016** | **0.826 ± 0.021** | **17.15 ± 1.57** |

### 5.3 EFFECTIVENESS OF PAIRWISE REPRESENTATION

To assess the efficacy of our pairwise encoder module, we compare PAIRE and AVICI on a skeleton prediction task. Note that, to eliminate any potential bias arising from differences in model size and further demonstrate our superior performance, we employ a 6-layer node feature encoder within the PAIRE framework in all experiments. Consequently, the PAIRE model comprises 2.8 million trainable parameters, while the AVICI model contains 3.2 million parameters. The experimental results on SBM random graphs are presented in Figure 3 and others are provided in Appendix. Our findings reveal that the PAIRE method consistently outperforms the AVICI approach across all datasets, despite having a smaller model size, which validates the effectiveness of the proposed pairwise encoder module.

### 5.4 EFFECTIVENESS OF LEARNING IDENTIFIABLE STRUCTURES

As discussed in Section 4.3, PAIRE focuses on learning identifiable causal structures rather than directly learning the adjacency matrix. Experimental results in Table 5 provide empirical support for its superiority. We compare two models with identical network structures but different learning targets: one model predicts the adjacency matrix, while the other predicts the skeleton and v-tensor together. Consistently, the models that predict the skeleton and v-tensor demonstrate higher accuracy in their predictions compared to the alternative model. This observation further strengthens our argument regarding the necessity and benefits of learning identifiable causal structures to improve overall performance. It is important to note that this comparison serves as an ablation study, highlighting the necessity of learning identifiable causal structures. PAIRE is specifically designed to excel at predicting the skeleton and v-structure due to its ability to capture pairwise representations.

## 6 CONCLUSION

We propose PAIRE, a novel DNN-based SCL approach. It incorporates a unique pairwise encoder module with a unidirectional attention layer designed to encode essential causal information. We utilize a skeleton matrix and a v-tensor as outputs, representing the Markov Equivalence Class (MEC) to address inconsistencies related to identifiability. Through extensive experiments conducted on both synthetic and real-world datasets, we demonstrate the superiority of our PAIRE framework over other DNN-based SCL methods.

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

## A1 THEORETICAL GUARANTEE

In this section, we delve into the theoretical analysis concerning the asymptotic correctness of our proposed model with respect to the sample size. Sec. A1.1 lays out the essential definitions and assumptions pertinent to the problem under study. Following this, from Sec. A1.2 to A1.3, we rigorously demonstrate the asymptotic correctness of the neural network model. Finally, in Sec. A1.4, we engage in a detailed discussion about the practical advantages and superiority of neural network models.

### A1.1 DEFINITIONS AND ASSUMPTIONS

As outlined in Sec. 3, a Causal Graphical Model is defined by a joint probability distribution $P$ over $d$ random variables $X_1, X_2, \cdots, X_d$, and a DAG $G$ with $d$ vertices representing the $d$ variables. An observational dataset $D$ consists of $n$ records and $d$ columns, which represents $n$ instances drawn i.i.d. from $P$. In this work, we assume causal sufficiency:

**Assumption A1.1** (Causal Sufficiency). *There are no latent common causes of any of the variables in the graph.*

Moreover, we assume the data distribution $P$ is Markovian to the DAG $G$:

**Assumption A1.2** (Markov Factorization Property). *Given a joint probability distribution $P$ and a DAG $G$, $P$ is said to satisfy Markov factorization property w.r.t. $G$ if $P := P(X_1, X_2, \cdots, X_d) = \prod_{i=1}^{d} P\left(X_i \mid \mathrm{pa}_i^G\right)$, where $\mathrm{pa}_i^G$ is the parent set of $X_i$ in $G$.*

It is noteworthy that the Markov factorization property is equivalent to the Global Markov Property (GMP) (Lauritzen, 1996), which is

**Definition A1.1** (Global Markov Property (GMP)). *$P$ is said to satisfy GMP (or Markovian) w.r.t. a DAG $G$ if $X \perp_G Y|Z \Rightarrow X \perp Y|Z$. Here $\perp_G$ denotes d-separation, and $\perp$ denotes statistical independence.*

GMP indicates that any d-separation in graph $G$ implies conditional independence in distribution $P$. We further assume that $P$ is faithful to $G$ by

**Assumption A1.3** (Faithfulness). *Distribution $P$ is faithful w.r.t. a DAG $G$ if $X \perp Y|Z \Rightarrow X \perp_G Y|Z$.*

**Definition A1.2** (Canonical Assumption). *We say our settings satisfy the canonical assumption if the Assumptions A1.1 - A1.3 are all satisfied.*

We restate the definitions of skeletons, Unshielded Triples (UTs) and v-strucutres as follows.

**Definition A1.3** (Skeleton). *A skeleton $E$ defined over the data distribution $P$ is an undirected graph where an edge exists between $X_i$ and $X_j$ if and only if $X_i$ and $X_j$ are always dependent in $P$, i.e., $\forall Z \subseteq \{X_1, X_2, \cdots, X_d\} \setminus \{X_i, X_j\}$, we have $X_i \not\perp X_j|Z$.*

Under our assumptions, the skeleton is the same as the corresponding undirected graph of $G$ Spirtes et al. (2000).

**Definition A1.4** (Unshielded Triples (UTs) and V-structures). *A triple of variables $X, T, Y$ is an Unshield Triple (UT) denoted as $\langle X, T, Y \rangle$, if $X$ and $Y$ are both adjacent to $T$ but not adjacent to each other in the DAG $G$ or the corresponding skeleton. It becomes a v-structure denoted as $X \to T \leftarrow Y$, if the directions of the edges are from $X$ and $Y$ to $T$ in $G$.*

We introduce the definition of separation set as:

**Definition A1.5** (Separation Set). *For a node pair $X_i$ and $X_j$, a vertex set $Z$ is a separation set if $X_i \perp X_j|Z$. Under faithfulness assumption, a separation set $Z$ is a subset of variables within the vicinity that d-separates $X_i$ and $X_j$.*

Finally, we assume a neural network can be used as a universal approximator in our settings.

**Assumption A1.4** (Universal Approximation Capability). *A neural network model can be trained to approximate a function under our settings with arbitrary accuracy.*

### A1.2 SKELETON LEARNING

In this section, we prove the asymptotic correctness of neural networks on the skeleton prediction task by constructing a perfect model and then approximating it with neural networks. For the sake of convenience and brevity in description, we define the skeleton predictor as follows.

**Definition A1.6** (Skeleton Predictor). *Given observational data $D$, a skeleton predictor is a predicate function with domain as observational data $D$ and predicts the adjacency between each pair of the vertices.*

Now we restate the Remark from Ma et al. (2022) as the following proposition. It proves the existence of a perfect skeleton predictor by viewing the skeleton prediction step of PC (Spirtes et al., 2000) as a skeleton predictor, which is proved to be sound and complete.

**Proposition A1.1** (Existence of a Perfect Skeleton Predictor). *There exists a skeleton predictor that always yields the correct skeleton with sufficient samples in $D$.*

*Proof.* We construct a skeleton predictor $SP$ consisting of two parts by viewing PC (Spirtes et al., 2000) as a skeleton predictor. In the first part, it extracts a pairwise feature $\boldsymbol{x}_{ij}$ for each pair of nodes $X_i$ and $X_j$:

$$\boldsymbol{x}_{ij} = \min_{Z \subseteq V \setminus \{X_i, X_j\}} \left\{ X_i \sim X_j \mid Z \right\}, \tag{1}$$

where $\left\{ X_i \sim X_j \mid Z \right\} \in [0, 1]$ is a scalar value that measures the conditional dependency between $X_i$ and $X_j$ given a vertex subset $Z$. Consequently, $\boldsymbol{x}_{ij} > 0$ indicates the persistent dependency between the two nodes.

In the second part, it predicts the adjacency based on $\boldsymbol{x}_{ij}$:

$$\left( X_i, X_j \right) = \begin{cases} 1 \text{ (adjacent)} & \boldsymbol{x}_{ij} \neq 0 \\ 0 \text{ (non-adjacent)} & \boldsymbol{x}_{ij} = 0 \end{cases} \tag{2}$$

Now we prove that $SP$ always yields the correct skeleton by proving the absence of false positive predictions and false negative predictions. Here, false positive prediction denotes $SP$ predicts a non-adjacent node pair as adjacent and false negative predictions denote $SP$ predicts an adjacent node pair as non-adjacent.

- **False Positive.** Suppose $X_i, X_j$ are non-adjacent. Under the Markovian assumption, there exists a set of nodes $Z$ such that $\left\{ X_i \sim X_j \mid Z \right\} = 0$ and hence $\boldsymbol{x}_{ij} = 0$. According to Equation (2), $SP$ will always predicts them as non-adjacent.

- **False Negative.** Suppose $X_i, X_j$ are adjacent. Under the faithfulness assumption, for any $Z \in V \setminus \{X_i, X_j\}$, $\left\{ X_i \sim X_j \mid Z \right\} > 0$, which implies $\boldsymbol{x}_{ij} > 0$. Therefore, $SP$ always predicts them as adjacent.

Therefore, $SP$ never yields any false positive predictions or false negative predictions under the Markovian assumption and faithfulness assumption, i.e., it always yields the correct skeleton. $\square$

With the existence of a perfect skeleton predictor, we prove the correctness of neural network models with sufficient samples under our assumptions.

**Theorem A1.1.** *Under the canonical assumption and the assumption that neural network can be used as a universal approximator (Assumption A1.4), there exists a neural network model that always predicts the correct skeleton with sufficient samples in $D$.*

*Proof.* From Proposition A1.1, there exists a perfect skeleton predictor that predicts the correct skeleton. Thus, according to the Assumption A1.4, a neural network model can be trained to approximate the perfect skeleton prediction hence predicts the correct skeleton. $\square$

### A1.3 ORIENTATION LEARNING

Similarly to the overall thought process in Sec. A1.2, in this section we prove the asymptotic correctness of neural networks on the v-structure prediction task by constructing a perfect model and then approximating it with neural networks.

**Definition A1.7** (V-structure Predictor). *Given observational data $D$ with sufficient samples from a BN with vertices $V = \{X_1, \ldots, X_p\}$, a v-structure predictor is a predicate function with domain as observational data $D$ and predicts existence of the v-structure for each unshielded triple.*

The following proposition proves the existence of a perfect v-structure predictor by viewing the orientation step of PC (Spirtes et al., 2000) as a v-structure predictor.

**Proposition A1.2** (Existence of a Perfect V-structure Predictor). *Under the Markov assumption and faithfulness assumption, there exists skeleton predictor that always yields the correct skeleton.*

*Proof.* We construct a v-structure predictor $VP$ consisting of two parts by viewing PC (Spirtes et al., 2000) as a v-structure predictor. In the first part, it extracts a feature $z_{ijk}$ for each UT $\langle X_i, X_k, X_j \rangle$:

$$z_{ijk} = \frac{\left| \{ (X_k, Z) | \{ X_i \sim X_j | Z \} = 0 \wedge X_k \in Z \} \right|}{\left| \{ Z | \{ X_i \sim X_j | Z \} = 0 \} \right|}, \tag{3}$$

□

where $\{ X_i \sim X_j \mid Z \} \in [0, 1]$ is a scalar value that measures the conditional dependency between $X_i$ and $X_j$ given a vertex subset $Z$, and $| \cdot |$ represents the cardinality of a set. Note that the denominator is always positive because the separation set of a UT always exists (See Lemma 4.1 in Dai et al. (2023)). Intuitively, $z_{ijk}$ represents the proportion of supsets of $X_i$ and $X_j$ that include $X_k$.

In the second part, it predicts the v-structures based on $z_{ijk}$:

$$\langle X_i, X_k, X_j \rangle = \begin{cases} 0 \text{ (not v-structure)} & z_{ijk} \neq 0 \\ 1 \text{ (v-structure)} & z_{ijk} = 0 \end{cases} \tag{4}$$

Now we prove that $VP$ always yields the correct predictions of v-structures. According to Theorem 5.1 on p.410 of Spirtes et al. (2000), assuming faithfulness and sufficient samples, if a UT $\langle X_i, X_k, X_j \rangle$ is a v-structure, then $X_k$ does not belong to any separation sets of $(X_i, X_j)$; if a UT $\langle X_i, X_k, X_j \rangle$ is not a v-structure, then $X_k$ belongs to every separation sets of $(X_i, X_j)$. Therefore, we have $z_{ijk} = 0$ if and only if $X_k$ is not in any separation set of $X_i$ and $X_j$, i.e., $\langle X_i, X_k, X_j \rangle$ is a v-structure.

With the existence of a perfect v-structure predictor, we prove the correctness of neural network models with sufficient samples under our assumptions.

**Theorem A1.2.** *Under the canonical assumption and the assumption that neural network can be used as a universal approximator (Assumption A1.4), there exists a neural network model that always predicts the correct v-structures with sufficient samples in $D$.*

*Proof.* From Proposition A1.1, there exists a perfect skeleton predictor that predicts the correct v-structures. Thus, according to the Assumption A1.4, a neural network model can be trained to approximate the perfect v-structure predictions hence predicts the correct v-structures. □

### A1.4 DISCUSSION

In the sections above, we prove the asymptotic correctness of neural network models by constructing theoretically perfect predictors. These predictors both consist of two parts: feature extractors providing features $x_{ij}$ and $z_{ijk}$, and final predictors of adjacency and v-structures. Even though they have a theoretical guarantee of the correctness with sufficient samples, it is noteworthy that they are hard to be applied practically. For example, to obtain $x_{ij}$ in Equation (1), we need to calculate the conditional dependency between $X_i$ and $X_j$ given every vertex subset $Z \subseteq V \setminus \{ X_i, X_j \}$. Leaving

aside the fact that the number of $Z$s itself presents factorial complexity, the main issue is that when $Z$ is relatively large, due to the curse of dimensionality, it becomes challenging to find sufficient samples to calculate the conditional dependency. This difficulty significantly hampers the ability to apply the constructed prefect predictors in practical scenarios.

Some existing methods can be interpreted as constructing more practical predictors. Majority-PC (MPC) (Colombo et al., 2014) achieves better performance on finite samples by modifying Equation 4 as:

$$\langle X_i, X_k, X_j \rangle = \begin{cases} 0 \text{ (not v-structure)} & z_{ijk} > 0.5 \\ 1 \text{ (v-structure)} & z_{ijk} \le 0.5 \end{cases} \tag{5}$$

Due to its more complex classification mechanism, it achieves better performance empirically. However, from the machine learning perspective, features from both the PC and MPC predictors are relatively simple. As supervised causal learning methods, ML4S (Ma et al., 2022) and ML4C (Dai et al., 2023) provide more systematic featurizations by manual feature engineering and utilization of powerful machine learning models for classification. While these methods show enhanced practical efficacy, their manual feature engineering processes are complex. In our paper, we utilize neural networks as universal approximators for learning the prediction of identifiable causal structures. It not only simplifies the procedure but also potentially uncovers more nuanced and complex patterns within the data that manual methods might overlook. It is noteworthy that the benefits of supervised causal learning using neural networks are also discussed elsewhere, as mentioned in CSIvA (Ke et al., 2023).

## A2   ILLUSTRATION OF THE CASE STUDY IN SEC. 4.3

Figure A4 presents an illustration for the case study of the Bernoulli-sampling adjacency matrix approach in Sec. 4.3. It clearly shows that observational data with the two different parametrized forms follow the same joint distribution:

$$P([X, Y, T]) = \mathcal{N}\left([0, 0, 0], \begin{bmatrix} 1 & 1 & 1 \\ 1 & 3 & 2 \\ 1 & 2 & 2 \end{bmatrix}\right). \tag{6}$$

Therefore, the observational datasets coming from the two DAGs are inherently indistinguishable.

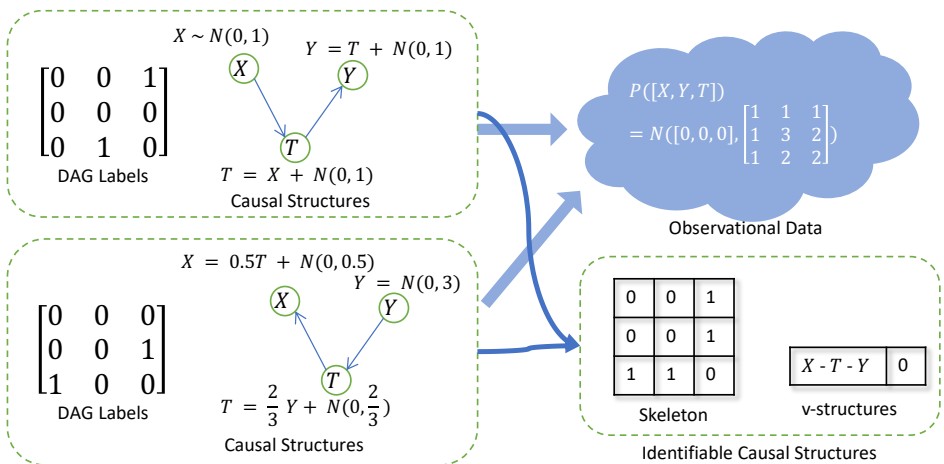

Figure A4: The problem setting to emphasize the limitations of the Bernoulli-sampling adjacency matrix approach. *Best viewed in color.*

## A3   EXPERIMENTAL SETTINGS

**Metrics.**   In all tables, ± indicates that the mean value and maximum deviation of three runs with different random seeds are reported. In the field of skeleton prediction tasks, the F1 score has

emerged as a widely adopted metric due to its ability to effectively balance precision and recall (Ding et al., 2020; Ma et al., 2022). This metric provides a comprehensive evaluation of the model's performance, particularly in cases where the data distribution is imbalanced. Accuracy, another commonly used metric, offers a direct measure of the proportion of misclassified edges within the graph. It can also be interpreted as a normalized version of the Structural Hamming Distance (SHD), which has gained popularity in recent years (Ma et al., 2022; Lorch et al., 2022; Ke et al., 2023).

Considering that deep learning models typically output probabilities rather than discrete labels, the Area Under the Receiver Operating Characteristic Curve (AUC) and the Area Under the Precision-Recall Curve (AUPRC) are also employed as more robust metrics. These metrics take into account all possible decision thresholds, providing a comprehensive evaluation of the model's performance across various operating points.

For the CPDAG prediction task, accuracy is used as a comparison metric, which measures the ratio of misclassified edges in the predicted CPDAG. Following the previous paper (Dai et al., 2023), the F1-scores calculated for identifiable edges and v-structures are also provided for a more comprehensive comparison.

**Baselines.** To demonstrate the effectiveness and superiority of the proposed framework, several strong baselines representing multiple categories are selected for comparison. These baselines include:

1. PC: A classic constraint-based causal discovery algorithm based on conditional independence tests. The version with parallelized optimization is selected (Le et al., 2016).
2. GES: A classic score-based greedy equivalence search algorithm (Chickering, 2002).
3. NOTEARS: A gradient-based algorithm for linear data models (Zheng et al., 2018).
4. DAG-GNN: A continuous optimization algorithm based on graph neural networks (Yu et al., 2019).
5. AVICI: A powerful deep learning-based supervised causal learning method (Lorch et al., 2022).
6. NOTEARS-MLP: A gradient-based algorithm for non-linear data models (Zheng et al., 2018).
7. GOLEM: A more efficient version of NOTEARS (Ng et al., 2020).
8. GraNDAG: A gradient-based algorithm using neural network modeling for non-linear additive noise data (Lachapelle et al., 2020).

The implementation from gCastle (Zhang et al., 2021) is utilized for the first four baselines. Note that the CSIvA model (Ke et al., 2023) is also a closely related method, but it is not compared due to the unavailability of its relevant codes and its requirement for interventional data as input. The original AVICI model (Lorch et al., 2022) does not support discrete data. Therefore, we use an embedding layer to replace its first linear layer when using AVICI on discrete data. All classic algorithms are run on an AMD EPYC 7V13 CPU, and DNN-based methods are run on an Nvidia A100 GPU. Our codes can be accessed at https://anonymous.4open.science/r/paire-C05D.

**Synthetic Data.** We randomly generate random graphs from multiple random graph models. For continuous data, following previous work (Lorch et al., 2022), Erdős-Rényi (ER) and Scale-free (SF) are utilized as the training graph distribution $p(G)$. The degree of training graphs in our experiments varies randomly among 1, 2, and 3. For testing graph distributions, Watts-Strogatz (WS) and Stochastic Block Model (SBM) are used, with parameters consistent with those in the previous paper (Lorch et al., 2022). All synthetic graphs for continuous data contain 30 nodes. The lattice dimension of Watts-Strogatz (WS) graphs is sampled from $\{2, 3\}$, yielding an average degree of about $4.92$. The average degrees of Stochastic Block Model (SBM) graphs are set at 2, following the settings in the aforementioned paper. For discrete data, 11-node graphs are used. SF is utilized as the training graph distribution $p(G)$ and ER is used for testing. The synthetic training data is generated in real-time, and the training process does not use the same data repeatedly. All synthetic test datasets contain 100 graphs, and the average values of the metrics on the 100 graphs are reported to comprehensively reflect the performance.

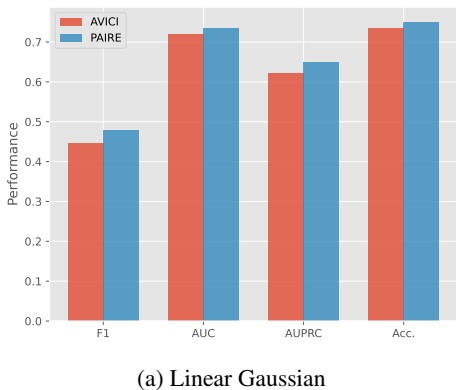
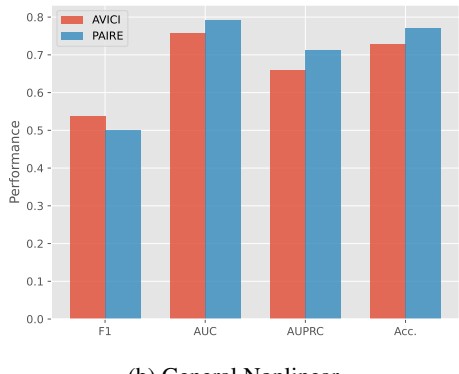

|               |               |
| :-----------: | :-----------: |
| (a) Linear Gaussian | (b) General Nonlinear |

Figure A5: Compairson between AVICI and PAIRE of Skeleton prediction task on WS Graphs.

For the forward sampling process from graph to continuous data, both the linear Gaussian mechanism and general nonlinear mechanism are applied. Concretely, the Random Fourier Function mechanism is used for the general nonlinear data following the previous paper (Lorch et al., 2022). In synthesizing discrete datasets, the Bernoulli distribution is used following previous papers (Dai et al., 2023; Ma et al., 2022).

**Post-processing.** Although our method theoretically guarantees asymptotic correctness, in practice, conflicts in predicted v-structures might occasionally occur in practice. Therefore, in the post-processing stage, we apply a straightforward heuristic to resolve the potential conflicts among predicted v-structures following previous work (Dai et al., 2023). After that, we use an improved version of Meek rules (Meek, 1995a; Tsagris, 2019) to obtain other identifiable edges without introducing extra cycles. Combining the skeleton from the skeleton predictor model with all identifiable edge directions, we get the final output of the CPDAG.

## A4 EXTRA EXPERIMENTAL RESULTS

Table A6: Comparison of skeleton prediction task on ER random graphs of discrete data.

| Method | F1 | AUC | AUPRC | Accuracy |
| :----- | :----: | :----: | :----: | :------: |
| AVICI | 0.833 | 0.961 | 0.925 | 0.914 |
| PAIRE | **0.862** | **0.976** | **0.952** | **0.921** |

**More Comparisons on Continuous Data.** Table A7 - A8 presents additionally comparison between PAIRE and more baseline methods on the WS dataset for both skeleton prediction task and CPDAG prediction task. PAIRE consistently demonstrates superior performance in comparison with these methods. Figure A5 presents an experimental comparison between AVICI and PAIRE on WS random graphs for skeleton prediction. These results reinforce the analysis in Sec. 5.3 and demonstrate the effectiveness of the proposed pairwise encoder module.

Figure A6 illustrates the test performance trends of the v-structure prediction model on SBM and WS random graphs during the training process. In this model, the feature extractor $FE$ is fine-tuned from the skeleton prediction model. The performance increases rapidly and achieves a relatively high level after just a few initial epochs. This suggests that the v-structure task is relatively straightforward, and the pre-trained pairwise features from the skeleton prediction model are both effective and generalizable.

**More Comparisons on Synthetic Discrete Data.** Since neural network model outputs range from 0 to 1 as probabilities rather than single predictions, AUC and AUPRC are more appropriate metrics

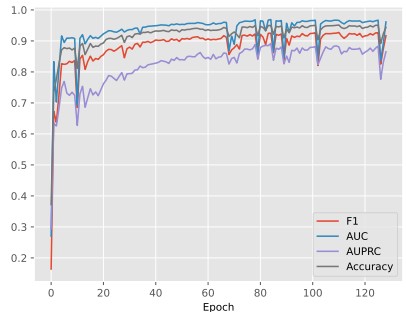

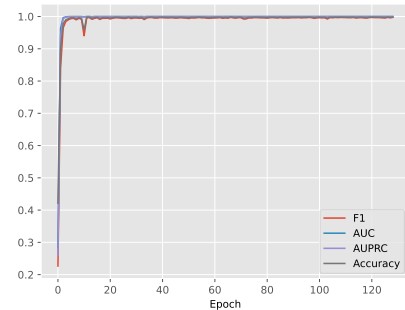

(a) Variation trends of test performance on WS graph.

(b) Variation trends of test performance on SBM graph.

Figure A6: Variation trends of the test performance of v-structure prediction model on WS and SBM random graphs during training.

Table A7: More comparison of skeleton prediction results on linear Gaussian data and WS graphs.

| Method | Skeleton F1 | Skeleton AUC |
|---|---|---|
| GRANDAG | 0.163 | 0.502 |
| NOTEARS-MLP | 0.256 | 0.513 |
| GOLEM | 0.293 | 0.539 |
| PAIRE | **0.479 ± 0.015** | **0.750 ± 0.003** |

for comparing DNN-based SCL methods. The comparison between AVICI and PAIRE for the skeleton prediction task on discrete data is provided in Table A6. By adjusting the classification threshold of DNN-based SCL methods, we can also compare them with traditional methods. These results are presented in Table A9. It is evident that DNN-based SCL methods outperform their counterparts, with PAIRE consistently achieving the best performance under these conditions.

**More Results on Sachs.** We present the comparison on F1 score and accuracy on the Sachs dataset in Figure A7. The results demonstrate that DNN-based SCL methods consistently outperform classical approaches, thereby confirming their effectiveness and superiority in this context.

**Training Data Diversity and Model Generalization.** We present experimental evidence that highlights the significant contribution of training data diversity to the model's generalization capabilities, even when applied to out-of-distribution (OOD) datasets. To illustrate this, we trained one PAIRE model on a combined dataset of both SF and ER, and another solely on the SF dataset. The comparative performance of these models is detailed in Table A10. The model trained on the combined ER and SF datasets exhibited markedly better per-

Table A9: Comparison of skeleton prediction task on ER random graphs of discrete data.

| Method | F1 | Accuracy |
|---|---|---|
| PC | 0.822 | 0.830 |
| GES | 0.821 | 0.818 |
| NOTEARS | 0.164 | 0.747 |
| AVICI | 0.833 | 0.914 |
| PAIRE | **0.862** | **0.921** |

Table A8: More comparison of CPDAG prediction results on linear Gaussian data and WS graphs.

| Method | V-structure F1 | Identfiable edges F1 | SHD |
|---|---|---|---|
| GRANDAG | 0.116 | 0.115 | 169.48 |
| NOTEARS-MLP | 0.128 | 0.126 | 192.59 |
| GOLEM | 0.158 | 0.191 | 172.63 |
| PAIRE | **0.298 ± 0.076** | **0.370 ± 0.062** | **116.797 ± 7.253** |

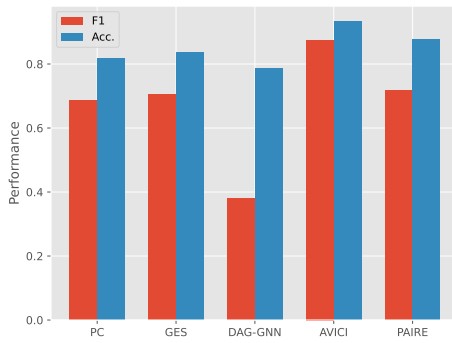

Figure A7: Skeleton prediction results on the Sachs dataset.

Table A10: Comparison of PAIRE models with different training data diversity on skeleton prediction.

(a) Model trained on both ER and SF

| Test Dataset | F1 | AUC | AUPRC | Accuracy |
|---|---|---|---|---|
| WS | 0.3631 | 0.7056 | 0.6061 | 0.7329 |
| SBM | 0.7811 | 0.9675 | 0.8809 | 0.9483 |
| ER | 0.8069 | 0.9603 | 0.8918 | 0.9473 |
| SF | 0.8473 | 0.9845 | 0.9355 | 0.9551 |

(b) Model trained on SF

| Test Dataset | F1 | AUC | AUPRC | Accuracy |
|---|---|---|---|---|
| WS | 0.4011 | 0.6300 | 0.4609 | 0.6353 |
| SBM | 0.6427 | 0.9165 | 0.7287 | 0.9087 |
| ER | 0.6706 | 0.9042 | 0.7393 | 0.9080 |
| SF | 0.8783 | 0.9886 | 0.9529 | 0.9611 |

formance, not only on the ER dataset but also on the other two OOD datasets, with only a marginal decrease in performance on the SF dataset. These findings suggest that enhancing the diversity of the training data correspondingly improves the model's ability to generalize and maintain robust performance across novel OOD datasets.

**Varying Amount of Training Graphs.** We present an analysis of how varying the amount of the training graphs influences performance on the skeleton prediction task. The results, depicted in Figure A8, illustrate a clear trend: model performance improves in tandem with the expansion of the training dataset. This trend underscores the potential of our method to achieve even greater accuracy given a more extensive dataset.

**Varying Sample Size.** We assessed PAIRE across various quantities of observational samples per graph during testing (100, 200, ..., 1000). The outcomes for both the skeleton prediction task and the CPDAG prediction task are depicted in Figure A9. It is evident that the model's performance enhances with the augmentation of sample size. These consistent upward trends suggest that PAIRE exhibits stability and is not overly sensitive to changes in sample size.

**Varying Edge Density.** We evaluate PAIRE over a range of edge densities in the test graphs, utilizing the SBM dataset, as it allows for the direct setting of average edge densities. The findings are presented in Figure A10. It's apparent that the task is becomes more difficult as edge densities increase. However, the performance decline is not abrupt, indicating that PAIRE's performance remains relatively stable across various edge densities, thereby confirming its versatility.

**Acyclicity.** We provide an empirical evidence supporting of the rarity of cycles in the final predictions. The experimental data presented in Table A11 corroborates that cycles are infrequently observed in the final predicted CPDAGs.

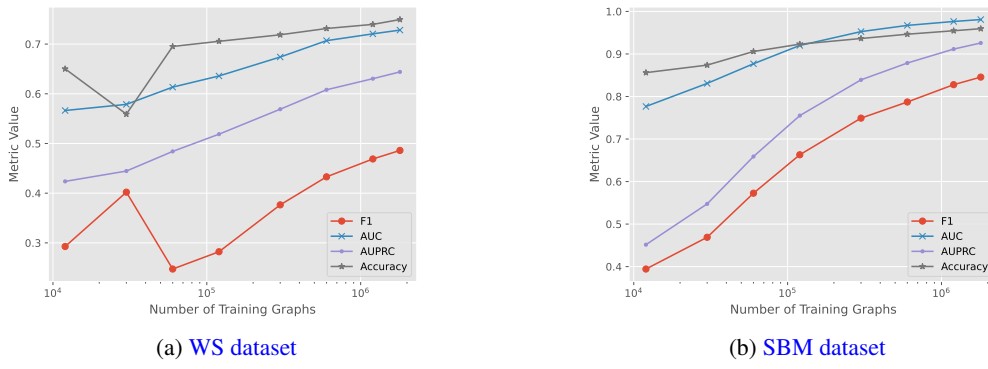

(a) WS dataset

(b) SBM dataset

Figure A8: Model performance with varying amount of training graphs.

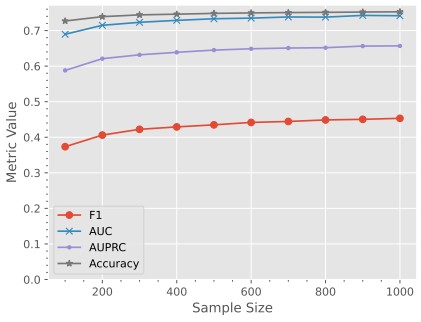

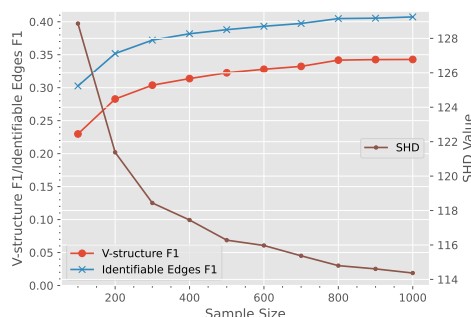

(a) Variation trends of skeleton predicton task performance on WS graph with varying sample sizes.

(b) Variation trends of CPDAG predicton task performance on WS graph with varying sample sizes.

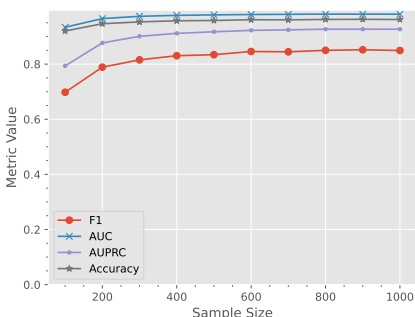

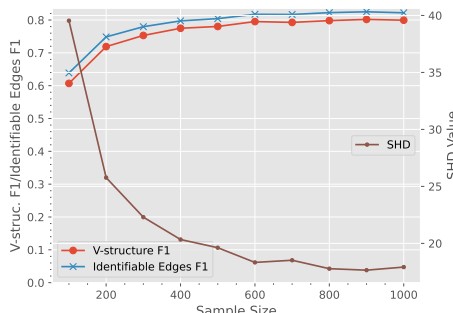

(c) Variation trends of skeleton predicton task performance on SBM graph with varying sample sizes.

(d) Variation trends of CPDAG predicton task performance on SBM graph with varying sample sizes.

Figure A9: Variation trends of performance with varying sample sizes.

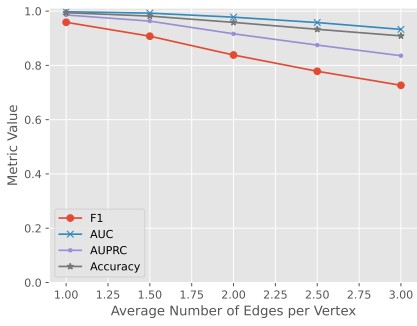
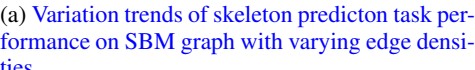
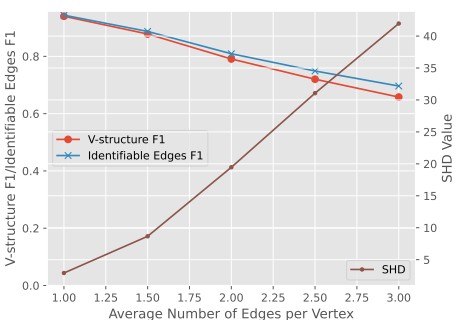

(a) Variation trends of skeleton predicton task performance on SBM graph with varying edge densities.

(b) Variation trends of CPDAG predicton task performance on SBM graph with varying edge densities.

Figure A10: Variation trends of performance with varying edge densities.

Table A11: Count of cycles in the final CPDAG predictions.

| Dataset | WS | SBM |
|---|---|---|
| Rate of Graphs with Cycles | $0.66 \pm 0.66\%$ | $0.00 \pm 0.00\%$ |

**Generality on Testing Graph Sizes.** We offer an analytical perspective on the performance of the PAIRE model when applied to larger WS graphs. It is important to highlight that the models were initially trained on graphs comprising 30 vertices, positioning this task within an out-of-distribution setting in terms of graph size. To establish a point of reference, we have included results from the PC algorithm as a baseline comparison. These findings can be examined in Table A12. Despite the OOD conditions, PAIRE maintains robust performance, reinforcing its scalability and the model's general applicability across varying graph sizes.

Table A12: Performance comparison with varying amounts of graph sizes.

| Metric | F1 Score | | | V-structure-F1 | | | Identifiable-Edges F1 | | |
|---|---|---|---|---|---|---|---|---|---|
| Graph Size | 50 | 70 | 100 | 50 | 70 | 100 | 50 | 70 | 100 |
| PC | 0.1767 | 0.1484 | 0.1060 | 0.0635 | 0.0504 | 0.0366 | 0.0699 | 0.0559 | 0.0403 |
| PAIRE | 0.4156 | 0.3738 | 0.2834 | 0.3494 | 0.3070 | 0.2261 | 0.3793 | 0.3369 | 0.2479 |

