# OpenReview forum: "Learning Identifiable Causal Structures with Pairwise Representation"
_ICLR.cc/2024/Conference — Submitted to ICLR 2024_

### Official Review · Reviewer_JJmt · 2023-10-29

**Soundness:** 3 good
**Presentation:** 3 good
**Contribution:** 3 good
**Rating:** 5
**Confidence:** 2

**Summary:**

This paper employs deep neural network, more precisely transformer-based architecture for supervised causal structure learning. The proposed method learns identifiable causal structures from discrete and continuous data.
Since both node features and pairwise features are used while training, the model learns the internal and external relationships of pairs of nodes. The proposed method learns the skeleton and v-structures separately which ensures the persistent dependency and orientation asymmetry. As a result,  the identifiability consistency of the Markov Equivalence class is ensured as well.

**Strengths:**

Here I provide some strengths of this paper.
* The authors described two issues of the closest baseline precisely and in detail. This would motivate the readers to dig deep into the paper and search for the solution.
* The paper is well written. The text in this paper indicates that the authors have good theoretical knowledge about the problem.
* I appreciate the authors for providing more precise details as “Remarks” at different parts of the paper.
* The authors included different baselines in their experiments which illustrate the performance of their proposed method compared to others.

**Weaknesses:**

Here I provide some weaknesses of this paper.

[Section 1]
* “Each entry Gij is sampled independently from Aij”- this can be paraphrased as “each entry Gij is sampled from Aij independently”
* The authors mentioned that non-zero probability of P_{XT}(1 - P_{TY}) resulting in the output of a v-structure   X -> T <- Y: this should be more intuitive.
* The last few sentences of the “Essential causal information” paragraph should be explained in more simple words.
* It is not clear what the authors meant by “second-order tensor” and “third-order tensor”.
* Some sentences of the paper should be simplified as those will help readers better understand the concepts.

[Section 2]
* The concept of “manual feature engineering” is not precise.
* “Permutation invariance across samples and permutation equivariance across variables” are not explained properly.

[Section 4.1]
* The authors mentioned about “standard forward sampling P(D|G)”. This should be more precise.

[Section 4.2]
* Figure 2 caption can contain some more details about the module.
* The concepts of “Internal and external information” should be explained more precisely.
* The authors should provide a brief description of attention operations for novice readers. Anyhow, the authors should explain more intuitively what value, key, and query mean for a pair of nodes.
* The authors mentioned that “It naturally maintains permutation equivariance across both the variable dimension and the data dimension”. The reason behind that is not explained properly.
* It is hard to follow how this architecture is contributing to causal learning or discovering the relations between a pair of variables.
* Different components of the pairwise encoder module can be annotated with some numbers and be used while describing about them.
* It is unclear how the node feature encoder is used. What did the authors mean by applying attention layers over “either the observation dimension or the node dimension alternately”? And how does that maintain permutation equivariance?

[Section 4.3]
* The authors mentioned “Considering a simulator that generates DAGs only from two DAGs with the same proportion”- this should be explained more precisely.
* The authors mentioned some potential conflicts among v-structures and heuristics to resolve that. No details about these were discussed.

[Section 5.1]:
* More details should be provided about Erdos-R´enyi (ER), Scale-Free (SF) mechanisms, Watts-Strogatz (WS), and Stochastic Block Model (SBM)  at least in the appendix.
* For the experiments, the authors should provide more information. such as the number of nodes, edge density, number of samples, etc. in the main paper.
[Section 5.2]
* It is not explained properly how the trained mode on SF graphs was used to discover the Sachs causal graph.


[Major concerns]
* The authors mentioned in the appendix that all their synthetic graphs contain 30 nodes and Sachs contains 11 nodes. This does not show if their algorithm is suitable for discovering larger graphs. The author should show their algorithm performance on larger graphs (with more than 100 nodes) which can be found in the Bnlearn repository (https://www.bnlearn.com/bnrepository/)

**Questions:**

[Section 1]:
* How is PAIRE different from their closest baseline AVICI in terms of methodology? It appears to me that they also used transformer-based architecture.

* How does the proposed algorithm adapt to a dataset with higher-dimensional variables? For example, if each sample contains d variables each having k dimensions, i.e., the dataset matrix will be D \in R^{n x d x k}.

[Section 4.1]:
* What did the authors mean by standard forward sampling P(D|G)? What is the assumed data-generating process?

[Section 4.2]
* What does the node feature refer to?
* What is the theoretical guarantee that this structure will always learn correct causal relations among variables given any datasets?
* The authors mentioned that they used “unidirectional attention operation”. What would be the alternative and what information that might represent for causal learning?


[Section 5]
* We know deep learning approaches take a lot of data to train on the other hand we do not generally have a large amount of real-world data for a specific causal graph. How will the proposed method adapt to those situations?
* Did the authors apply the mentioned heuristic and the Meek rules for baseline algorithms as well?
* What are the reasons behind classical algorithms’ worse performance for the chosen datasets?
* How did the authors apply the trained model on the Sachs graph? How was the Sachs observational data used? These details should also be made precise.

[Major concerns]
* How are the authors making sure that there would be no cycles during skeleton learning? Their mentioned baseline AVICI seemed to optimize for acyclicity while it is not clear how the authors are ensuring that.
* If a model is trained on synthetic data that follows a specific distribution, that trained model might not perform well on any new dataset with a different data distribution due to any bias toward previous training datasets. In those situations, the existing constraint-based causal discovery algorithms might perform better. How does the proposed method in this paper deal with this issue?
* Can the authors show their performance with varying edge density and sample size?

I am willing to increase the score if the major concerns are properly dealt with.

---

> ### Author Response · Authors · 2023-11-18
> **Response to Reviewer JJmt (1/3)**
>
> Thank you for your constructive comments and suggestions, which have been immensely helpful in enhancing our paper. We have meticulously integrated your feedback into the revised manuscript. Below, we first restate your comments, followed by our detailed responses to each point.
>
> > Reviewer JJmt wonders how PAIRE handles cycles.
>
> Firstly, in the task of skeleton learning, the predicted skeleton is an undirected graph, thereby precluding the presence of cycles. We assume the reviewer questions how to handle cycles in CPDAG prediction. Theoretically, our updated analysis in Appendix Sec. A1 shows that PAIRE is theoretically proven to be asymptotically correct and consistency. This implies that cycles are never predicted under ideal cases. Practically, the occurrence of cycles is also rare in our predictions and PAIRE performs well in our experiments.
>
> > Reviewer JJmt wonders the transferability of PAIRE (The test distribution is different with the training distribution).
>
> The PAIRE model, trained on randomly generated datasets, is specifically designed to adeptly capture causal relations through our pairwise representations. As detailed in the Appendix Sec. A3, in our experimental evaluations, we primarily focus on out-of-distribution (OOD) settings, where the test distribution is distinct from the training distribution. As the experimental results presented in Sec. 5 indicate, PAIRE exhibits strong generality under this OOD setting. It is worth to note that the transferability of Supervised Causal Learning is also evidenced in previous work (Dai et al., 2023).
>
>
> > Reviewer JJmt wonders more experiments about the performance with varying edge density and sample size.
>
> **Varying Sample Size.** We assessed PAIRE across various quantities of observational samples per graph during testing (100, 200, ..., 1000). Owing to space limitations, we showcase four key metrics in the table below: AUC and ACC for skeleton prediction, along with V-structure F1 and Identifiable Edges F1 for CPDAG prediction. The visualized figures for all metrics are provided in the revised Appendix Sec. A4. The model's performance enhances with the augmentation of sample size. These consistent upward trends suggest that PAIRE exhibits stability and is not overly sensitive to changes in sample size.
>
>
> - WS Dataset
>
> |       | 1000  | 900   | 800   | 700   | 600   | 500   | 400   | 300   | 200   | 100   |
> |-------|-------|-------|-------|-------|-------|-------|-------|-------|-------|-------|
> | AUC   | 0.742 | 0.743 | 0.738 | 0.738 | 0.735 | 0.733 | 0.729 | 0.723 | 0.715 | 0.690 |
> | ACC   | 0.753 | 0.752 | 0.751 | 0.751 | 0.750 | 0.748 | 0.746 | 0.744 | 0.739 | 0.727 |
> | V-structure F1   | 0.343 | 0.343 | 0.342 | 0.332 | 0.328 | 0.323 | 0.314 | 0.304 | 0.283 | 0.230 |
> | Identifiable Edges F1   | 0.407 | 0.405 | 0.405 | 0.397 | 0.393 | 0.388 | 0.382 | 0.372 | 0.352 | 0.303 |
>
> - SBM Dataset
>
> |       | 1000  | 900   | 800   | 700   | 600   | 500   | 400   | 300   | 200   | 100   |
> |-------|-------|-------|-------|-------|-------|-------|-------|-------|-------|-------|
> | AUC   | 0.981 | 0.981 | 0.981 | 0.981 | 0.980 | 0.979 | 0.977 | 0.974 | 0.965 | 0.934 |
> | ACC   | 0.962 | 0.963 | 0.962 | 0.961 | 0.961 | 0.958 | 0.957 | 0.953 | 0.946 | 0.920 |
> |  V-structure F1  | 0.800 | 0.802 | 0.798 | 0.793 | 0.795 | 0.780 | 0.775 | 0.753 | 0.719 | 0.607 |
> | Identifiable Edges F1   | 0.822 | 0.825 | 0.823 | 0.817 | 0.818 | 0.804 | 0.797 | 0.780 | 0.749 | 0.639 |
>
>
> **Varying Edge Density.** We evaluated PAIRE over a range of edge densities in the test graphs, utilizing the SBM dataset, as it allows for the direct setting of average edge densities. Owing to space limitations, we showcase four key metrics in the table below: AUC and Acc. for skeleton prediction, along with V-structure F1 and Identifiable Edges F1 for CPDAG prediction. The visualized figures for all metrics are provided in the revised Appendix Sec. A4.  It's apparent that the task is becomes more difficult as edge densities increase. However, the performance decline is not abrupt, indicating that PAIRE's performance remains relatively stable across various edge densities, thereby confirming its versatility.
>
>
> | Avg. Edges per Vertex | 1.0   | 1.5   | 2.0   | 2.5   | 3.0   |
> |-------|-------|-------|-------|-------|-------|
> | AUC   | 0.998 | 0.993 | 0.978 | 0.958 | 0.933 |
> | ACC   | 0.994 | 0.982 | 0.959 | 0.933 | 0.909 |
> | V-structure F1  | 0.940 | 0.878 | 0.791 | 0.720 | 0.658 |
> | Identifiable Edges F1 | 0.944 | 0.887 | 0.809 | 0.749 | 0.696 |

---

> > ### Comment · Reviewer_JJmt · 2023-11-20
> > **Some concerns**
> >
> > I thank the authors for their responses. I have the following questions.
> >
> > 1. Can the authors please explain with a little more detail which part of their algorithm (without referring to the paper) is making sure that there will be no cycles and how?
> >
> > 2. Does the gCastle repository apply meek rules for the baselines? If they do not but the authors apply meek rules for their proposed method then it would be an unfair comparison.
> >
> > 3. They have assumptions such as causal sufficiency, Markovian property, faithfulness, and that neural networks function as a universal approximation. The authors suggest that the proposed algorithm performs better in the test distribution which might be arbitrarily different from the training distribution and the prediction is done without any information about how the test data was generated. Aren't the authors putting a lot of faith in the trained neural networks without any theoretical guarantee?

---

> > > ### Author Response · Authors · 2023-11-22
> > > **Further Response to Reviewer JJmt (2/2)**
> > >
> > > > Reviewer JJmt wonders how PAIRE handles diverse test distributions which might be different from training distributions.
> > >
> > > Reviewer t5Qz raises a similar question about how to obtain sufficient training data in terms of both amount and diversity. We'd like to provide a same response below:
> > >
> > > We would like to first clarify the **two types of data sufficiency** essential for PAIRE: sampling data sufficiency and training data sufficiency. Our theoretical analysis, detailed in Appendix Sec. A1, reveals the existence of perfect deep learning models that always predict correct results with sampling data sufficiency. Nevertheless, the actual development of such an ideal model depends on **training data sufficiency, including both data amount and data diversity**. While this problem is undoubtedly complex and difficult, we **offer additional empirical evidence and progress below**, underscoring the promising prospects of SCL methods.
> > >
> > > 1. **How to obtain sufficient training data.** As mentioned in Sec. 1, the training data for SCL methods can be readily obtained via synthetic generation. Different with most of DNN tasks, we are able to generate infinite observations for a given causal graph with an approximate free cost. We could also generate infinite random graphs with current random graph generators. Therefore, it offers the significant advantage of the acquisition of training data, allowing for extensive and sufficient training of the models. The effectiveness of this paradigm has been verified in previous SCL methods (Dai et al., 2023; Ke et al., 2023; Ma et al., 2022).
> > >
> > > 2. The generality of DNN models is a fundamental problem in deep learning and we currently cannot definitively guarantee that our training data is "sufficient" to approximate the theoretically perfect model. However, the experimental results presented below provide evidence that both an increase in the amount of training data and its diversity contribute to enhance model performance. This underscores PAIRE's capacity and potential to achieve superior results with a larger and more diverse training dataset.
> > >     - **Training data amount.** The table below illustrates the increase of PAIRE's performance as the amount of training data increases. For a visual representation of the trend, please refer to the revised Appendix, Figure A8.
> > > | Number of Training Graphs | F1     | AUC    | AUPRC  | Accuracy |
> > > | ------------------------- | ------ | ------ | ------ | -------- |
> > > | 12000                     | 0.3943 | 0.7766 | 0.4516 | 0.8561   |
> > > | 36000                     | 0.5018 | 0.8425 | 0.5767 | 0.8785   |
> > > | 60000                     | 0.5726 | 0.8770 | 0.6588 | 0.9058   |
> > > | 120000                    | 0.6630 | 0.9200 | 0.7551 | 0.9230   |
> > > | 300000                    | 0.7492 | 0.9526 | 0.8391 | 0.9363   |
> > > | 600000                    | 0.7871 | 0.9672 | 0.8787 | 0.9464   |
> > > | 1200000                   | 0.8278 | 0.9764 | 0.9115 | 0.9546   |
> > > | 1800000                   | 0.8456 | 0.9809 | 0.9258 | 0.9594   |
> > >     - **Training data Diversity.** The table below illustrates the performance of PAIRE trained on a combination of ER and SF graphs, and only SF graphs, with the same amount. The model trained on the combined ER and SF datasets exhibited markedly better performance, not only on the ER dataset but also on the other two OOD datasets, with only a marginal decrease in performance on the SF dataset. It verifies that enhancing the diversity of the training data improves the model generality.
> > >
> > >
> > > **Model trained on ER + SF**
> > >
> > > | Dataset | F1     | AUC    | AUPRC  | ACC    |
> > > |-------|--------|--------|--------|--------|
> > > | WS    | 0.3631 | 0.7056 | 0.6061 | 0.7329 |
> > > | SBM   | 0.7811 | 0.9675 | 0.8809 | 0.9483 |
> > > | ER    | 0.8069 | 0.9603 | 0.8918 | 0.9473 |
> > > | SF    | 0.8473 | 0.9845 | 0.9355 | 0.9551 |
> > >
> > > **Model trained on SF**
> > >
> > > | Dataset | F1     | AUC    | AUPRC  | ACC    |
> > > | ------- | ------ | ------ | ------ | ------ |
> > > | WS      | 0.4011 | 0.6300 | 0.4609 | 0.6353 |
> > > | SBM     | 0.6427 | 0.9165 | 0.7287 | 0.9087 |
> > > | ER      | 0.6706 | 0.9042 | 0.7393 | 0.9080 |
> > > | SF      | 0.8783 | 0.9886 | 0.9529 | 0.9611 |

---

> ### Author Response · Authors · 2023-11-18
> **Response to Reviewer JJmt (2/3)**
>
> > Reviewer JJmt wonders the theoretical guarantee of PAIRE.
>
> Reviewer 4F7j has expressed a concern similar to yours and we'd like to provide a same response:
> - **Theoretical Guarantee.** As outlined in Sec. 3 and further elaborated upon in the revised Appendix Sec. A1, our model is based on three assumptions which are commonly used in the causal discovery domain: causal sufficiency, Markovian property, and faithfulness. These assumptions are widely recognized and employed in related work (Ma et al., 2022)(Dai et al., 2023). Furthermore, by assuming that the neural network functions as a universal approximator, we provide a theoretical foundation for the asymptotic correctness w.r.t. sample size and consistency of our method in Appendix Sec. A1.
> - **Discussion on Practicability.** Even though our method has a theoretical guarantee of the correctness with infinite samples, it has distinct advantages as a kind of Supervised Causal Learning (SCL) method under the finite samples conditions. For example, traditional constraint-based methods usually suffer from the unreliablity of conditional independence test in practice. Additionally, search-based methods tend to underperform because of the exponential growth of the search space, which is also mentioned in related work (Ke et al., 2023). However, SCL methods can learn from supervised training data about how to make predictions from practical finite samples, which makes the causal structure learning more robust and general. We have elaborated the contents above in the revised Appendix Sec. A1.4.
>
> > Reviewer JJmt wonders more experiments on larger graphs.
>
> As evidenced in Sec. 5, we have verified that PAIRE outperform other methods on the 30-node graphs, which is a typical setting of the graph size in the literature (Ke et al., 2023)(Lorch et al., 2022). Currently we do not have the capability for experiments on the larger graphs due to the limited computational resource, and the improvement of the scalability is left as interesting future work.
>
> > Reviewer JJmt wonders the difference between PAIRE and AVICI.
>
> - As we summarize in Sec. 1, we design pairwise representations to capture essential causal information, and propose to use a skeleton matrix with a v-tensor as the learning target. Our ablation study shown in Sec. 5.3 - 5.4 verify the effectiveness of these innovations.
>
> > Reviewer JJmt wonders how PAIRE handles the higher-dimensional variables.
>
> - In the standard framework of causal discovery, each variable constitutes a unidimensional probability distribution. When dealing with high-dimensional variables, the graph representation and the corresponding causal structures are unclear. We leave it as interesting future work.
>
> > Reviewer JJmt wonders more details about the standard forward sampling.
>
> As detailed in Appendix Sec. A3, we apply both the linear Gaussian mechanism and a general nonlinear mechanism for continuous data generation. The Random Fourier Function mechanism is used for the general nonlinear data following the previous paper (Lorch et al., 2022). For discrete data, the Bernoulli distribution is employed. We have revised Appendix Sec. A3 to further clarify it.
>
> > Reviewer JJmt wonders what the node feature refers to in Sec. 4.2.
> - As elaborated in the final paragraph of Sec. 4.2, the node features refer to the features of each individual variable. These are derived from the node feature encoder module within our model. We have revised Appendix Sec. 4.2 to further clarify it.
>
> > Reviewer JJmt wonders more reasons of using the unidirectional attention operation.
>
> The unidirectional attention operation is motivated by two key insights:
>   - As highlighted in Sec. 1, it is crucial to incorporate both pairwise and node information to effectively capture essential causal information, such as persistent dependency and orientation asymmetry. To this end, unidirectional attention is employed to utilize information from both pairwise features and node features and represent the essential causal informtion.
>   - In terms of computational complexity, unidirectional attention is less demanding than an alternative self-attention operation involving pairwise features alone. Consequently, this approach also serves to reduce the overall computational costs.
>
> We have further clarified it in the revised Sec. 4.2.
>
> > Reviewer JJmt wonders how to obtain sufficient high-quality training data for our DNN model.
>
> - As mentioned in Sec. 1, the training data for SCL methods can be readily obtained via synthetic generation. Different with some other tasks, we are able to generate infinite observations for a given causal graph with an approximate free cost. Therefore, it offers the significant advantage of the acquisition of training data, allowing for extensive and sufficient training of the models. The effectiveness of this paradigm has been verified in previous SCL methods (Dai et al., 2023; Ke et al., 2023; Ma et al., 2022).

---

> ### Author Response · Authors · 2023-11-18
> **Response to Reviewer JJmt (3/3)**
>
> > Reviewer JJmt wonders whether the mentioned heuristic and Meek rules are applied for the baseline algorithms.
>
> As detailed in Appendix Sec. A3, we have employed the implementation provided by the gCastle repository for multiple baseline methods. In the case of AVICI, the heuristic and Meek rules are not applicable, as AVICI directly predicts the Directed Acyclic Graph (DAG).
>
> > Reviewer JJmt wonders why classical algorithms perform worse on the experiments.
>
> We adhere to the default settings provided in the gCastle library, and detailedly check their predictions to ensure correctness. The relevant codes and datasets will be made available for release.
>
>
> > Reviewer JJmt wonders more details about the experiments on Sachs dataset.
>
> As demonstrated in Sec. 5.2, we trained the PAIRE model on randomly generated scale-free (SF) graphs and then used it to predict the causal graph based on observational data from the Sachs dataset. The observational data, sourced from the Bnlearn repository, is readily compatible and can be directly processed by our trained PAIRE model.
>
>
> > Reviewer JJmt has suggestions on paper writing.
>
> Thanks for your careful reading and suggestions. We have revised the paper according to your suggestions.

---

> ### Author Response · Authors · 2023-11-22
> **Further Response to Reviewer JJmt (1/2)**
>
> We thank you for your further response and we answer the remaining questions below.
>
> > Reviewer JJmt wonders more details to make sure PAIRE will not yield cycles.
>
> To substantiate the chaim that cycles are infrequent in the final predictions, we quantified the occurrence of cycles within the predictions:
>
> | Dataset | WS           | SBM          |
> |---------|--------------|--------------|
> | Rate of Graphs with Cycles | $0.66 \pm 0.66$(%) | $0.00 \pm 0.00$(%) |
>
> Furthermore, to address the occasional emergence of cycles, we propose a straightforward algorithm to ensure acyclicity:
>
> ```
> For each detected cycle composed of edges from multiple v-structures:
>     Identify the v-structure with the lowest prediction score.
>     Remove the identified v-structure.
> End For
> ```
>
> This procedure iteratively eliminates the least confident v-structures, thereby resolving cycles effectively.
>
> We have revised the Appendix Sec. A4 to further clarify the acyclicity of the PAIRE predictions.
>
>
> > Reviewer JJmt wonders more about how does baselines in gCastle apply meek rules.
>
> The application of Meek rules is in line with the standard procedures for general constraint-based causal discovery methods, as also depicted in Fig. 2 and Sec. 2.1 of [2].  Among all of the baselines, only PC belongs to constraint-based methods, and the Meek rules are not applicable to other non-contraint-based methods. We check the codes and find that the PC implementation in gCastle applys Meek rules at https://github.com/huawei-noah/trustworthyAI/blob/master/gcastle/castle/algorithms/pc/pc.py#L203.
>
> [2] Yu, K., Li, J., & Liu, L. (2016). A review on algorithms for constraint-based causal discovery. arXiv preprint arXiv:1611.03977. Avaiable at: https://arxiv.org/abs/1611.03977v1

---

### Official Review · Reviewer_mhH7 · 2023-10-30

**Soundness:** 3 good
**Presentation:** 3 good
**Contribution:** 3 good
**Rating:** 6
**Confidence:** 4

**Summary:**

This paper provided a novel approach for supervised causal learning that relies on a pairwise encoder module. The encoder module uses multi-head attention on a pairwise concatenated tensor. The author(s) claim that this improves the model’s ability to capture the pairwise relationship between data and graph. Experiments on predicting the skeleton and V-structure demonstrate superior performance against SOTA.

**Strengths:**

This paper successfully pointed out the issues with current approaches. The novel workflow of PAIRE is very interesting. The definitions and writing are generally clear, and experimental results are convincing.

**Weaknesses:**

For the experiments, the author(s) discussed the number of parameters in Section 5.3, but not in Section 5.1. I wonder if the number of parameters across methods in Table 2 are comparable. It would be nice to have a FLOPs comparison between methods, since the pairwise decoder is more computationally intensive then regular self-attention. In addition, the purpose of Section 5.3 is unclear, since as far as I understand, it essentially restates the data in Table 1 and adds two evaluation metrics.

Minor issues: The figure captions could use more explanation as they should be self-contained. Also, the phrase ‘pairwise relationship’ is not clear, especially in Section 4.1 where it is unclear if the pairs are (D,G) or pair of nodes.

**Questions:**

- What are the number of parameters for methods in Table 1 and Table 2? Are there FLOPs comparisons?
- What is the intuition behind using pairwise feature only for the Query, and what is the unidirectional attention mentioned in Section 4.2?

---

> ### Author Response · Authors · 2023-11-18
> **Response to Reviewer mhH7**
>
> Thank you for your constructive comments and suggestions, which have been immensely helpful in enhancing our paper. We have meticulously integrated your feedback into the revised manuscript. Below, we first restate your comments, followed by our detailed responses to each point.
> > Reviewer mhH7 wonders more comparison on the number of parameters and FLOPs.
>
> As outlined in Sec. 5.3, our experiments with PAIRE utilize a 6-layer node feature encoder, amounting to 2.8 million trainable parameters. In comparison, the AVICI method comprises 3.2 million trainable parameters. It's important to note that this comparison of parameter count is not directly applicable to other baseline methods, as they do not employ SCL techniques. Regarding the FLOPs, PAIRE requires 38M FLOPs while AVICI requires 33M FLOPs in processing one batch of graphs. Although PAIRE's computational cost is marginally higher, it remains within a similar range.
>
> > Reviewer mhH7 wonders more about the unidirectional attention.
>
> The unidirectional attention means only using the pairwise features as the Query, and the node features as both Key and Value. This approach is motivated by two key insights:
>   - As highlighted in Sec. 1, it is crucial to incorporate both pairwise and node information to effectively capture essential causal details, such as persistent dependency and orientation asymmetry. To this end, unidirectional attention is employed to utilize information from both pairwise features and node features.
>   - In terms of computational complexity, unidirectional attention is less demanding than a self-attention operation involving pairwise features alone. Consequently, this approach also serves to reduce the overall computational costs.
>
> We have further clarified it in the revised Sec. 4.2.
>
> > Reviewer mhH7 suggests to add some explanation in certain parts of the paper.
>
> Thanks for your suggestion. We have revised the figure captions and the expression in Sec. 4.1 accrodingly.

---

### Official Review · Reviewer_t5Qz · 2023-10-31

**Soundness:** 2 fair
**Presentation:** 3 good
**Contribution:** 2 fair
**Rating:** 3
**Confidence:** 3

**Summary:**

This paper improves a transformer-based supervised causal discovery method, named CSIvA. Authors first identify two limitations of CSIvA, and then propose PAIRE that can model both internal and external relationships of variable pairs, to address these limitations.

**Strengths:**

- Identifying two limitations of an existing method an then incorporating skeleton and and v-structure information to improve the method.
- Presentation is good.

**Weaknesses:**

- Hard to have further useful information about the inferred result.
- The method claimed to resolve identifiability inconsistency, but there is no such guarantee even with infinite observed data.
- Proposed method is not scalable (see below) and is limited to small problems.
- Experiments are not convincing: lacking setup details, lacking completive benchmark method, and some better results in the literature were not reported.

**Questions:**

### Major concerns

- The proposed method is mostly algorithmic and lacks guarantees or further useful information of the inferred results.
  - The skeleton and v-structure are all inferred by a trained model, and there is no consistency guarantee, because we do not know how good the trained model is. In fact, in the experiment, authors only use 100 simulated graphs with 30-nodes in the training data. By no means this model would work well for all causal discovery settings.
  - And we can hardly get other useful information beyond the inferred graph. To be specific, for a constraint-based method, one can conclude that the absence of an edge between $X$ and $Y$ is because they are (conditionally) independent. For a score based method, the obtained graph has the best score (under some score function and possibly other assumptions) among all searched graphs. However, for the proposed method, there is no such information at all.
  - The pairwise representation used in this method may be beneficial, but it takes $d^2h$ -dim where $d$ is the number of nodes. If we have a 100-node problem and each original feature has $h=4$ dim, this would be $40,000$-dim feature. I think this design would limit the scalability of the problem. Please clarify.
- The experiments are not convincing:
  - lacking some important setup details: e.g., how are "general nonlinear data" generated  and what nonlinearity is used? what is the average degree of the graph? Also, why GES cannot finish in 24 hours for a 30-node problem?
  - lacking completive benchmark method: there are many other methods that are empirically good, like GraN-DAG, RL-BIC, NOTEARS-ML. Even though they return only a DAG, one can convert this DAG to its CPDAG and check how good the CPDAG is. Note that GOLEM has specially designed a score function for linear-Gaussian data with nonequal variances. Please also give a try.
  - important results were omitted: Existing works have better estimation results on Sachs data. For example, the method CSIvA has reported SHDs 5 and 6 on this dataset, while the proposed method has SHD=10. Of course, a method does not need to beat all the rest on all settings, but giving such results can be very informative for readers.

### Minor Concerns

- "Deep Neural Network (DNN) based SCL, which learns DNNs as causal models, has gained significant attention with its numerous advantages"--- I think here this type of methods is overclaimed; see the above of my perspective on such methods. Please revise.
- Since the paper considers transformer based NN model with sampling methods to generate a causal graph, some previous works earlier than CSIvA shall be discussed, as they also use transformer model and/or Bernoulli sampling , like below.
  - [1] causal discovery with reinforcement learning, ICLR 2020
  - [2]  Neural Causal Structure Discovery from Interventions

- I am not very convinced by the "Risk of Bernoulli-sampling adjacency matrix approach". The reasoning is that edge orientation is done by Bernoulli sampling and may cause as inconsistency problem. However, if we follow the same reasoning, the v-structure in the proposed method is also determined by "logit" and may yield a wrong decision as well.



Overall,  while the paper proposes an improved method over CSIvA , the technical contribution is weak, there is no guarantee or other useful information of inferred graph, and the experiments are not convincing. I recommend REJ for the current version.

---

> ### Author Response · Authors · 2023-11-18
> **Response to Reviewer t5Qz (1/2)**
>
> Thank you for your constructive comments and suggestions, which have been immensely helpful in enhancing our paper. We have meticulously integrated your feedback into the revised manuscript. Below, we first restate your comments, followed by our detailed responses to each point.
>
> > Reviewer t5Qz concerns PAIRE lacks theoretical consistency guarantee.
>
> Reviewer 4F7j has expressed a concern similar to yours and we'd like to provide a same response:
>
> - **Theoretical Guarantee.** As outlined in Sec. 3 and further elaborated upon in the revised Appendix Sec. A1, our model is based on three assumptions which are commonly used in the causal discovery domain: causal sufficiency, Markovian property, and faithfulness. These assumptions are widely recognized and employed in related work (Ma et al., 2022)(Dai et al., 2023). Furthermore, by assuming that the neural network functions as a universal approximator, we provide a theoretical foundation for the asymptotic correctness w.r.t. sample size and consistency of our method in Appendix Sec. A1.
> - **Discussion on Practicability.** Even though our method has a theoretical guarantee of the correctness with infinite samples, it has distinct advantages as a kind of Supervised Causal Learning (SCL) method under the finite samples conditions. For example, traditional constraint-based methods usually suffer from the unreliablity of conditional independence test in practice. Additionally, search-based methods tend to underperform because of the exponential growth of the search space, which is also mentioned in related work (Ke et al., 2023). However, SCL methods can learn from supervised training data about how to make predictions from practical finite samples, which makes the causal structure learning more robust and general. We have elaborated the contents above in the revised Appendix Sec. A1.4.
>
> > Reviewer t5Qz wonders more about the compairson with the listed methods.
>
> We present the comparison between PAIRE and the aforementioned methods on the WS dataset, following the default settings in the gCastle library.
>
> | Model       | Skeleton F1 | Skeleton AUC | V-struc. F1 | Identfiable edges F1 | SHD     |
> | ----------- | ----------- | ------------ | --------- | ----------------- | ------ |
> | GRANDAG     | 0.163      | 0.502       | 0.116    | 0.115            | 169.48 |
> | NOTEARS-MLP | 0.256      | 0.513       | 0.128    | 0.126            | 192.59 |
> | GOLEM       | 0.293      | 0.539       | 0.158    | 0.191            | 172.63 |
> |  PAIRE   |  0.479 +- 0.015   | 0.750 +- 0.003             |   0.298 +- 0.076     | 0.370 +- 0.062                  | 116.797 +- 7.253       |
>
> PAIRE consistently demonstrates superior performance in comparison with these methods. We have included the above results in the revised Appendix Sec. A4. Note that the RL-BIC method is time-consuming and fails to produce results within a reasonable timeframe.
>
> > Reviewer t5Qz suggests us to discuss two related papers [1, 2].
> > - [1] causal discovery with reinforcement learning, ICLR 2020
> > - [2] Neural Causal Structure Discovery from Interventions
>
> Our work is more related to CSIvA compared with the two mentioned papers. Paper [1] is a score-based method rather than a supervised causal learning method. It suggests to use Reinforcement Learning (RL) in the search process, employing a transformer network in the encoder and outputs Bernoulli distributions for the adjacency matrix. Unlike CSIvA (Ke et al., 2023), AVICI (Lorch et al., 2022), and our approach, their transformer network lacks sample-wise and node-wise attention mechanisms. We have add a relevant discussion in Sec. 2 in our revised manuscript. Paper [2] introduces a continuous optimization method named Structure Discovery from Interventions (SDI). Its setup is different with PAIRE as it integrates both observational and interventional data.
>
>
> > Reviewer t5Qz wonders why CSIvA reports a seemingly better result than PAIRE on Sachs.
>
> As noted in Appendix Sec. A3, additional data of single interventions on each variable is available under the CSIvA settings hence CSIvA predicts the DAG. Our PAIRE is designed to learn from observational data and predicts CPDAG. Therefore, a direct comparison is not meaningful. Moreover, as noted in Appendix Sec. A3, since the code for CSIvA has not been made publicly available, we are unable to modify it for a comparative analysis with PAIRE.

---

> ### Author Response · Authors · 2023-11-18
> **Response to Reviewer t5Qz (2/2)**
>
> > Reviewer t5Qz concerns 100 simulated graph is insufficient for PAIRE training.
>
> As noted in Sec. 4.1, both the training graph $G$ and the corresponding dataset matrix $D$ are generated in real-time. They are not reused during training. Consequently, this approach potentially provides an infinite amount of training data for the model. A typical PAIRE model is trained for 300 epochs, with each epoch comprising 6,000 training graphs. This results in a cumulative total of 1,800,000 training graphs throughout the training process. As we mentioned in Appendix Sec. A3, each synthetic test datasets contain 100 graphs. We have further clarified this in the revised Appendix Sec. A3.
>
>
> > Reviewer t5Qz wonders more details about data generation and average degree of graphs.
>
> Comprehensive details regarding our experimental setup can be found in Appendix Section A3. As discussed in the 'Synthetic Data' paragraph, we adopted the Random Fourier Function mechanism for generating general nonlinear data, in line with the approach used by Avici (Lorch et al. 2022). The degree of training graphs in our experiments varies randomly among 1, 2, and 3. For testing, the average degree of Watts-Strogatz (WS) graphs is about 4.9, while the average degree of Stochastic Block Model (SBM) graphs is 2, following the settings in the aforementioned study. We have revised the Appendix Sec. 3 to include the average degrees of the graphs.
>
> > Reviewer t5Qz wonders why GES failed on the WS dataset.
>
> GES aims to identify an optimal DAG based on a specific score function. This search process, however, can be time-intensive and encounter computational challenges. In our experiments, we utilized the default settings from the gCastle library. It indeed failed to produce results within 24 hours. This experience underscores the benefits of employing DNN-baed SCL: once trained, these models offer a significantly faster inference process.
>
>
> > Reviewer t5Qz concerns the scalability of PAIRE on larger graph.
>
> In the whole paper, we have revealed the significance and potential of pairwise representations, as they are crucial for capturing essential information such as persistent dependency and orientation asymmetry. As specifically evidenced in Sec. 5.3, we have proved that models incorporating pairwise representations outperform the original models on the 30-node problems, which is a typical setting of graph size in the literature (Ke et al., 2023)(Lorch et al., 2022). The improvement of the scalability on larger graphs is left as interesting future work.

---

> > ### Comment · Reviewer_t5Qz · 2023-11-21
> >
> > Thanks for your detailed response which addresses many of my concerns. Some remaining concerns are:
> >
> > - I checked the new theoretic result: "Under the canonical assumption and the assumption that neural network can be
> > used as a universal approximator (Assumption A1.4), there exists a neural network model that always predicts the correct skeleton with sufficient samples in D ". --- my understanding is that besides the infinite observed data of a specific problem, it assumes (1) universal approximation of NNs and (2) sufficiently diverse training data. I have further doubts (also somehow mentioned in previous  comments): how can you obtain such training data that are guaranteed to be "sufficient", in terms of both amount and diversity?
> >
> > - To clarify, based on the response, the current limitations of the proposed method are: 1) scalability (30-node is considered to be "small" in the current literature); 2) cannot handle interventional data.

---

> > > ### Author Response · Authors · 2023-11-22
> > > **Further Response to Reviewer t5Qz (1/2)**
> > >
> > > Thank you for getting back to us. We are glad that we addressed most of your concerns and that our explanations updated your evaluation of our work.
> > >
> > > > Reviewer t5Qz further wonders how to obtain sufficient training data in terms of both amount and diversity.
> > >
> > >
> > > We would like to first clarify the **two types of data sufficiency** essential for PAIRE: sampling data sufficiency and training data sufficiency. Our theoretical analysis, detailed in Appendix Sec. A1, reveals the existence of perfect deep learning models that always predict correct results with sampling data sufficiency. Nevertheless, the actual development of such an ideal model depends on **training data sufficiency, including both data amount and data diversity**. While this problem is undoubtedly complex and difficult, we **offer additional empirical evidence and progress below**, underscoring the promising prospects of SCL methods.
> > >
> > > 1. **How to obtain sufficient training data.** As mentioned in Sec. 1, the training data for SCL methods can be readily obtained via synthetic generation. Different with most of DNN tasks, we are able to generate infinite observations for a given causal graph with an approximate free cost. We could also generate infinite random graphs with current random graph generators. Therefore, it offers the significant advantage of the acquisition of training data, allowing for extensive and sufficient training of the models. The effectiveness of this paradigm has been verified in previous SCL methods (Dai et al., 2023; Ke et al., 2023; Ma et al., 2022).
> > >
> > > 2. The generality of DNN models is a fundamental problem in deep learning and we currently cannot definitively guarantee that our training data is "sufficient" to approximate the theoretically perfect model. However, the experimental results presented below provide evidence that both an increase in the amount of training data and its diversity contribute to enhance model performance. This underscores PAIRE's capacity and potential to achieve superior results with a larger and more diverse training dataset.
> > >     - **Training data amount.** The table below illustrates the increase of PAIRE's performance as the amount of training data increases. For a visual representation of the trend, please refer to the revised Appendix, Figure A8.
> > > | Number of Training Graphs | F1     | AUC    | AUPRC  | Accuracy |
> > > | ------------------------- | ------ | ------ | ------ | -------- |
> > > | 12000                     | 0.3943 | 0.7766 | 0.4516 | 0.8561   |
> > > | 36000                     | 0.5018 | 0.8425 | 0.5767 | 0.8785   |
> > > | 60000                     | 0.5726 | 0.8770 | 0.6588 | 0.9058   |
> > > | 120000                    | 0.6630 | 0.9200 | 0.7551 | 0.9230   |
> > > | 300000                    | 0.7492 | 0.9526 | 0.8391 | 0.9363   |
> > > | 600000                    | 0.7871 | 0.9672 | 0.8787 | 0.9464   |
> > > | 1200000                   | 0.8278 | 0.9764 | 0.9115 | 0.9546   |
> > > | 1800000                   | 0.8456 | 0.9809 | 0.9258 | 0.9594   |
> > >
> > >     - **Training data Diversity.** The table below illustrates the performance of PAIRE trained on a combination of ER and SF graphs, and only SF graphs, with the same amount. The model trained on the combined ER and SF datasets exhibited markedly better performance, not only on the ER dataset but also on the other two OOD datasets, with only a marginal decrease in performance on the SF dataset. It verifies that enhancing the diversity of the training data improves the model generality.
> > >
> > > **Model trained on ER + SF**
> > > | Dataset | F1     | AUC    | AUPRC  | ACC    |
> > > |-------|--------|--------|--------|--------|
> > > | WS    | 0.3631 | 0.7056 | 0.6061 | 0.7329 |
> > > | SBM   | 0.7811 | 0.9675 | 0.8809 | 0.9483 |
> > > | ER    | 0.8069 | 0.9603 | 0.8918 | 0.9473 |
> > > | SF    | 0.8473 | 0.9845 | 0.9355 | 0.9551 |
> > >
> > > **Model trained on SF**
> > > | Dataset | F1     | AUC    | AUPRC  | ACC    |
> > > | ------- | ------ | ------ | ------ | ------ |
> > > | WS      | 0.4011 | 0.6300 | 0.4609 | 0.6353 |
> > > | SBM     | 0.6427 | 0.9165 | 0.7287 | 0.9087 |
> > > | ER      | 0.6706 | 0.9042 | 0.7393 | 0.9080 |
> > > | SF      | 0.8783 | 0.9886 | 0.9529 | 0.9611 |

---

> ### Author Response · Authors · 2023-11-22
> **Further Response to Reviewer t5Qz (2/2)**
>
> > Reviewer t5Qz concerns the scalability of PAIRE on larger graphs.
>
> Even though we cannot train models on larger datasets due to the limitation of time and computational resouce currently, we present the performance of the PAIRE model when applied to larger WS graphs in the Table below. The models were initially trained on graphs comprising 30 vertices, positioning this task within an OOD setting in terms of graph size. To establish a point of reference, we also include results from the PC algorithm as a baseline comparison. Despite the OOD conditions, PAIRE maintains robust performance, verifying its scalability and the model’s general applicability across varying graph sizes.
>
>
> | Metrics     | F1 Score            |                    |                    |                    | V-structure-F1     |                    |                    |                    | Identifiable-Edges F1 |                    |                    |                    |
> |----------------|---------------------|--------------------|--------------------|--------------------|--------------------|--------------------|--------------------|--------------------|-----------------------|--------------------|--------------------|--------------------|
> | Size           | 50                  | 70                 | 80                 | 100                | 50                 | 70                 | 80                 | 100                | 50                    | 70                 | 80                 | 100                |
> | PAIRE    | 0.4156              | 0.3738             | 0.3262             | 0.2834             | 0.3494             | 0.3070             | 0.2638             | 0.2261             | 0.3793                | 0.3369             | 0.2912             | 0.2479             |
> | PC          | 0.1767              | 0.1484             | 0.1198             | 0.1060             | 0.0635             | 0.0504             | 0.0419             | 0.0366             | 0.0699                | 0.0559             | 0.0458             | 0.0403             |
>
>
> > Reviewer t5Qz concerns PAIRE cannot handle interventional data.
>
>
> Causal discovery from observational data is a well-established and extensively studied problem, which serves as the primary focus of our paper as shown in Sec. 1. Therefore, this paper does not target causal learning from interventional data.
>
> Nevertheless, extending PAIRE to support interventional data is an interesting future work. The two key components of PAIRE, i.e., pairwise representation and learning the identifiable causal structures, can both be extended to interventional data. When we can additionally access interventional data, the identifiable causal structure is up to an I-MEC (short for Interventional-Markov equivalence class) [1], which is a generalization of MEC. It is characterized by skeleton and v-structures as proved by Theorem 3.9 in [1]. Thus, identifiability is a unique characteristic of causal structure learning that we should respect.
>
> Regarding the pairwise representation, we present an ablation study of it in the simple setting of causal discovery from interventional data: We simulated single interventions with targets on all nodes separately so the i-MEC here is exactly the DAG. As shown in the table, the model with pairwise representation clearly outperforms the one without using pairwise representation. Further exploration of DNN-based SCL methods on interventional data is left as our future work.
>
> | Testset | Model                             | AUPRC | AUC   | F1    | Accuracy  |
> |---------|-----------------------------------|-------|-------|-------|-------|
> | WS      | Model without pairwise representation | 0.560 | 0.869 | 0.340 | 0.858 |
> | WS      | Model with pairwise representation    | 0.601 | 0.879 | 0.477 | 0.867 |
> | SBM     | Model without pairwise representation | 0.812 | 0.976 | 0.683 | 0.965 |
> | SBM     | Model with pairwise representation    | 0.862 | 0.982 | 0.758 | 0.971 |
>
>
> [1] Karren D. Yang, Abigail Katcoff, and Caroline Uhler. Characterizing and Learning Equivalence Classes of Causal DAGs under Interventions. In ICML, 2018.

---

### Official Review · Reviewer_4F7j · 2023-11-01

**Soundness:** 2 fair
**Presentation:** 2 fair
**Contribution:** 2 fair
**Rating:** 3
**Confidence:** 3

**Summary:**

This work endeavors to train a deep neural network using observational data and a ground truth graph, with the objective of generalizing its predictive capabilities to infer causal structures in new datasets. The proposed network features a distinctive pairwise encoder module, augmented by a unidirectional attention layer, and leverages a skeleton matrix in combination with a v-tensor, a third-order tensor representing v-structures, as the output. Empirical evidence supports the superior performance of PAIRE in comparison to other deep neural network-based Structural Causal Learning (SCL) methods.

**Strengths:**

My main concerns are as follows:

1) Theoretical guarantees: The problem setting appears unconventional to me. Specifically, I'm interested in understanding how the proposed method can theoretically establish its ability to identify true causal structures. From my understanding, the identifiability of general causal structures from observational data typically hinges on specific assumptions, such as model assumptions like linear non-Gaussian or additive noise models for nonlinear causal discovery. These assumptions play a crucial role in both the data generative process and the methods employed. How do the proposed networks align with such assumptions to ensure their capacity for identifiability? Although the proposed method primarily targets the identification of Markov equivalence classes, it is essential to understand how the proposed networks integrate independent tests into the training process.

2) The motivation also appears to have certain issues. For instance, the motivation that using the Bernoulli-sampling adjacency matrix as a learning target can produce inconsistent results raises concerns. To substantiate this, in Section 4.3.1, an example is provided involving two datasets generated by two distinct linear Gaussian models. In this context, I believe that the non-identifiability result is a consequence of model assumptions, specifically the fact that linear non-Gaussian models are not identifiable (with unequal variances), rather than being directly attributed to the use of Bernoulli sampling.

3) The experiments appear to be concentrated on specific non-identifiable scenarios, such as linear Gaussian models or general nonlinear models. It raises the question of why the authors have not included experiments in identifiable cases, like linear non-Gaussian models. I would like to recommend the addition of more experiments in these identifiable cases to demonstrate the advantages of the proposed method.

Overall, I think that significant improvements are necessary to enhance the clarity of theroblem setting and motivation.

**Weaknesses:**

See above

**Questions:**

See above

---

> ### Author Response · Authors · 2023-11-18
> **Response to Reviewer 4F7j**
>
> Thank you for your constructive comments and suggestions, which have been immensely helpful in enhancing our paper. We have meticulously integrated your feedback into the revised manuscript. Below, we first restate your comments, followed by our detailed responses to each point.
>
> > Reviewer 4F7j wonders our assumptions and more theoretical guarantee.
> - **Theoretical Guarantee.** As outlined in Sec. 3 and further elaborated upon in the revised Appendix Sec. A1, our model is based on three assumptions which are commonly used in the causal discovery domain: causal sufficiency, Markovian property, and faithfulness. These assumptions are widely recognized and employed in related work (Ma et al., 2022)(Dai et al., 2023). Furthermore, by assuming that the neural network functions as a universal approximator, we provide a theoretical foundation for the asymptotic correctness w.r.t. sample size and consistency of our method in Appendix Sec. A1.
> - **Discussion on Practicability.** Even though our method has a theoretical guarantee of the correctness with infinite samples, it has distinct advantages as a kind of Supervised Causal Learning (SCL) method under the finite samples conditions. For example, traditional constraint-based methods usually suffer from the unreliablity of conditional independence test in practice. Additionally, search-based methods tend to underperform because of the exponential growth of the search space, which is also mentioned in related work (Ke et al., 2023). However, SCL methods can learn from supervised training data about how to make predictions from practical finite samples, which makes the causal structure learning more robust and general. We have elaborated the contents above in the revised Appendix Sec. A1.4.
>
>
>
> > Reviewer 4F7j suggests to clarify the assumptions to criticize the Bernoulli-sampling adjancency matrix approach.
>
> We clarify that when the final prediction is up to a Markov Equivalence Class (MEC) rather than a fully identifiable DAG, the Bernoulli-sampling adjacency matrix approach may result in inconsistency for Unshielded Triples formed by non-identifiable edges. For instance, with linear Gaussian and discrete data, it is established that only the MEC consisting of skeleton and v-structure is identifiable. This Bernoulli-sampling adjancency matrix approach, as demonstrated in Sec. 1, leads to inconsistency in such cases. We have further clarified it in the revised Sec. 4.3.1.
>
> > Reviewer 4F7j wonders more experimental results under other settings, like linear non-Gaussian models.
>
>
> As shown in Sec. 5.1 - 5.2, we have conducted experiments in more general setups, such as general nonlinear and discrete data. We believe these settings are more general and difficult than the linear non-Gaussian models.

---

> ### Comment · Reviewer_4F7j · 2023-11-22
>
> Thanks for your detailed response.
>
> I align with Reviewer t5Qz in key concerns, particularly regarding the theoretical guarantees and the method's capacity to seamlessly integrate key assumptions, such as conditional independence, for identifiability. Unfortunately, after reviewing the responses to both Reviewer t5Qz's comments and mine, I have not received clear clarification.
>
> Further, from a modeling perspective, nonlinear models offer a broader scope than linear models. However, considering information content, nonlinear models provide richer information that can contribute to causal discovery. Therefore, it remains valuable to assess the performance of your method on standard linear non-Gaussian models.
>
> Given these considerations, I maintain my current rating.

---

> > ### Author Response · Authors · 2023-11-22
> > **Further Response to Reviewer 4F7j**
> >
> > Thank you for your feedback and we realize that your main concerns are highly aligned with Reviewer t5Qz. We think so far we have addressed most concerns of Reviewer t5Qz including 1) generality against training sufficiency 2) scalability 3) inteventional data handling. We'd like to clarify that the data sufficiency for PAIRE includes both sampling data sufficiency and training data sufficiency. Our theory in Appendix Sec. A1 provides the guarantee of model correctness under sampling data sufficiency. For analysis on the training data sufficiency, please refer to our *Further Reponse to Reviewer t5Qz (1/2)*.
> >
> > For concerns 2) and 3), please also refer to our *Further Response to Reviewer t5Qz (2/2)*.
> >
> > Regarding "seemlessly integrate key assumption", we do not fully understand your question. With our assumptions elaborated in Appendix Sec. A1.1, our work has a theoretical guarantee with respect to the sampling data sufficiency. If you have other confusion, please let us know.
> >
> > > Reviewer 4F7j suggests more experiments on nonlinear models, specially, linear non-Gaussian models.
> >
> > We are working on the experiments on this. Considering the deadline of the author response period, we cannot provide the result currently and we will update this result in the final version.

---

### Author Response · Authors · 2023-11-18
**General Response**

We extend our gratitude to the reviewers for their constructive feedback. We are pleased to note that our work has garnered **positive remarks from multiple respects**, including:

- Impact
    - "successfully pointed out the issues with current approaches." -- Reviewer mhH7
    - "...would motivate the readers to dig deep into the paper and search for the solution." -- Reviewer JJmt
- Novelty
    - "The novel workflow of PAIRE is very interesting." -- Reviewer mhH7
- Writing
   -  "Presentation is good." -- Reviewer t5Qz
    - "The paper is well written." -- Reviewer JJmt
    - "The definitions and writing are generally clear." -- Reviewer mhH7
- Completeness
    - "... providing more precise details as “Remarks” at different parts of the paper." -- Reviewer JJmt
- Experimental Evaluation:
   -  "experimental results are convincing." -- Reviewer mhH7
   -  "included different baselines in their experiments which illustrate the performance of their proposed method compared to others." --  Reviewer JJmt

In response to the feedback from Reviewers 4F7j, t5Qz, and JJmt regarding our theoretical guarantees, we have **included additional theoretical analysis about the asymptotic correctness w.r.t. sample size, along with a related discussion**, introduced in Sec. 4.3.2 and detailed in Appendix Sec. A1.

Following the suggestions by the reviewers, we provide **extra experimental results**:
- Comparison with more baselines, including GRANDAG, GOLEM, GraN-DAG, and RL-BIC. -- Reviewer t5Qz
- Performance study with varying edge density and sample size. -- Reviewer JJmt

We have carefully considered the writing suggestions provided by the reviewers:

-  Use more explanation in figure captions. -- Reviewer mhH7, Reviewer JJmt
-  More clearly describe pairwise relationship in Sec. 4.1. -- Reviewer mhH7
-  Annotate different components of the pairwise encoder module with some numbers. -- Reviewer JJmt
-  Other writing suggestions from reviewers.

We would like to express our gratitude to the reviewers once more, and we welcome the opportunity for further discussion.

---

### Author Response · Authors · 2023-11-22
**Further General Response**

We appreciate the reviewers' feedback and are pleased that our previous responses have resolved most of their concerns, including, but not limited to:
- Theoretical guarantee on the asymptotic correctness w.r.t. sample size
- Experimental comparison with extra baselines
- Generality on varying graph edge density and sample size

The current focus of the reviewers' concerns can be summarized as follows, and we have responded to each point:

- **Training data sufficiency**: We provide a relevant discussion and empirical evidence in both *Further Response to Reviewer t5Qz (1/2)* and *Further Response to Reviewer JJmt (2/2)*.
- **Generality on graph size scalability**: Additional experimental results confirming PAIRE's scalability with varying graph sizes are provided in the *Further Response to Reviewer t5Qz (2/2)*.

Some reviewers also wonder more about the acyclity of PAIRE predictions and the interventional data handling. These points are elaborated on in the *Further Response to Reviewer JJmt (1/2)* and *Further Response to Reviewer t5Qz (2/2)*, respectively.

A relatively minor yet pending item involves additional experiments on linear non-Gaussian data (Given that we have already provided experimental results on linear Gaussian and general nonlinear data). We are committed to conducting further experiments on this.

We would like to express our gratitude to the reviewers once more, and we welcome the opportunity for further discussion.

---

### Meta-Review · Area_Chair_Sgpn · 2023-12-06

**Metareview:**

A supervised causality learning method called PAIRE is proposed in order to address two limitations of the existing methods identified by the authors. The proposed model takes node features and feature interactions as inputs, and outputs both a skeleton matrix and a v-tensor to represent the MEC.

**Justification For Why Not Higher Score:**

Several reviewers raised concerns that the proposed method lacks not only rigorous theoretical results but also convincing experimental results.

**Justification For Why Not Lower Score:**

N/A

---

### Decision · Program_Chairs · 2024-01-16

Reject